# Engineering sulfonate group donor regeneration systems to boost biosynthesis of sulfated compounds

Ruirui Xu[1,2,3], Weijao Zhang[1,2,3], Xintong Xi[1,2,3], Jiamin Chen[1,2,3], Yang Wang ®[1,2,3], Guocheng Du ®[1,2,3], Jianghua Li[1,2,3], Jian Chen ®[1,2,3] & Zhen Kang ®[1,2,3] ✉

Sulfonation as one of the most important modification reactions in nature is essential for many biological macromolecules to function. Development of green sulfonate group donor regeneration systems to efficiently sulfonate compounds of interest is always attractive. Here, we design and engineer two different sulfonate group donor regeneration systems to boost the biosynthesis of sulfated compounds. First, we assemble three modules to construct a 3'-phosphoadenosine-5'-phosphosulfate (PAPS) regeneration system and demonstrate its applicability for living cells. After discovering adenosine 5'-phosphosulfate (APS) as another active sulfonate group donor, we engineer a more simplified APS regeneration system that couples specific sulfotransferase. Next, we develop a rapid indicating system for characterizing the activity of APS-mediated sulfotransferase to rapidly screen sulfotransferase variants with increased activity towards APS. Eventually, the active sulfonate group equivalent values of the APS regeneration systems towards trehalose and *p*-coumaric acid reach 3.26 and 4.03, respectively. The present PAPS and APS regeneration systems are environmentally friendly and applicable for scaling up the biomanufacturing of sulfated products.

Sulfonation reactions that catalyzed by diverse sulfotransferases[1,2] transfer a sulfonate group (-SO$_3$H) to a hydroxyl group in compounds and perform one of the most important modification reactions in nature[3,4]. Sulfonation is essential for bioactivation or detoxification[5,6] because it changed the biophysical properties of biomolecules, such as polysaccharides[7,8], proteins[9–11], hormones[12], and lipids[13]. Due to the diversified physiological functions and wide applications[14] of these sulfate-containing compounds, the biomanufacturing of these compounds is imperative. Moreover, 3'-phosphoadenosine-5'-phosphosulfate (PAPS) has been accepted as the unique active sulfonate group donor for all biological sulfonation reactions[15,16] and adequate supply of PAPS is a prerequisite for accomplishing efficient biosynthesis of these sulfated compounds.

As pioneering research, a rat aryl sulfotransferase IV (AST IV)-dependent PAPS regeneration system was creatively constructed with *p*-nitrophenyl sulfate (PNPS) as the primary sulfonate donor[17]. Thereafter, different AST IV enzymes have been investigated to optimize the PAPS regeneration system for rapid determination of sulfotransferase activity and enzymatic synthesis of chondroitin sulfate[18,19] and heparin[20]. In view of the high price of 3'-phosphoadenosine-5'-phosphate (PAP), PAPS was added for one-pot chemoenzymatic synthesis of heparin with the AST IV-dependent PAPS regeneration system[21–23]. However, accumulation of the toxic byproduct *p*-nitrophenol (PNP) and the comparatively high cost of substrate PNPS restrict their large-scale applications[24]. Consequently, many studies have recently concentrated on the biosynthesis of PAPS with adenosine 5'-triphosphate

[1]The Key Laboratory of Carbohydrate Chemistry and Biotechnology, Ministry of Education, School of Biotechnology, Jiangnan University, Wuxi 214122, China. [2]The Science Center for Future Foods, Jiangnan University, Wuxi 214122, China. [3]The Key Laboratory of Industrial Biotechnology, Ministry of Education, School of Biotechnology, Jiangnan University, Wuxi 214122, China. ✉e-mail: zkang@jiangnan.edu.cn

(ATP) as substrate[25–27]. To reuse the byproduct adenosine 5'-diphosphate (ADP) and increase the conversation rate, a high-energy compound phosphoenolpyruvate was introduced[28]. Recently, polyphosphate especially $polyP_6$ has been applied as phosphate donors to drive ATP regeneration for enzymatic biosynthesis of valuable compounds[29–31] because of its low price. In particular, after creating an artificial bifunctional ATP sulfurylase-APS (adenosine-5'-phosphosulfate) kinase (ASAK) and screening pyrophosphate (PPi) kinase, Xu and colleagues engineered efficient ATP regeneration systems with intrinsic byproduct PPi or supply of $polyP_6$ for equimolar conversion of PAPS from ATP[32]. Even so, large amounts of PAPS are still required for sufficient sulfonation while the accumulated PAP exhibits feedback inhibition on sulfotransferases[33,34]. Hence, it is preferable to engineer alternative green sulfonate group donor regeneration systems for biomanufacturing of sulfated products.

In this work, a de novo PAPS regeneration system is designed and constructed by integrating three modules for ATP biosynthesis from AMP, conversion of PAPS from ATP and regeneration of AMP from PAPS, in which specific sulfotransferase is coupled for sulfonating compounds of interest. We also demonstrate its applicability in living cells for biosynthesis of sulfated products. We discover and validate APS, the precursor of PAPS, is also an active sulfonate group donor. An indicating system is constructed for rapidly characterizing and screening sulfotransferase variants with higher activities towards APS. After engineering the PAPS/APS binding site 3'-PB (3'-phosphate-binding motif) region of sulfotransferases, the APS utilization efficiency is significantly improved. Eventually, an APS regeneration system with short route (three enzymes in total) is developed for sulfonation.

## Results

### Design of an in vitro PAPS regeneration system

Although the AST IV-dependent PAPS regeneration system has been used for enzymatic synthesis of sulfonated compounds[18,21,35], more green alternative PAPS regeneration systems are still imperative. Inspired by the studies on the widely distributed PAPS metabolism[36] (Fig. 1a), we designed a PAPS regeneration system by assembling three modules: the conversion of ATP from AMP, the biosynthesis of PAPS from ATP and the regeneration of AMP from PAPS, in which the inexpensive substrate sulfate and polyphosphate were used as initial raw materials (Fig. 1b). For Module I, polyphosphate kinases, especially the polyphosphokinase family 2 (PPK2) members that exhibit AMP and ADP phosphorylase activities[37] and apply inexpensive polyphosphate (polyP) chains as substrates[38] were selected for characterization. As shown in Fig. 1c, liquid chromatogram analysis showed that apart from $PPK2^{p.a-N}$, all the other polyphosphate kinases exhibited both AMP and ADP phosphorylase activities. In contrast, $PPK2^{s.e}$ (from *S. epidermidis*) showed the highest enzymatic activity and preferentially utilized AMP (Fig. 1d, Supplementary Fig. 1a). Additionally, the polymerization degree of polyphosphates for supporting AMP and ADP phosphorylases was also investigated, and the results suggested that the degree of polyphosphate polymerization for AMP phosphorylation should be higher than three while for ADP phosphorylation, the minimal polymerization degree was 2 (Supplementary Fig. 1b).

For Module II, the artificial bifunctional PAPS synthase ASAK that constructed in our previous work[32] was utilized. Considering the adverse effect of excessive polyphosphate $polyP_6$ on the enzyme activity of ASAK[32], the concentration of $polyP_6$ was optimized and determined to be 2.0 mM in the reaction system (Supplementary Fig. 1c). After coupling the sulfonate group transfer reaction, the produced byproduct PAP with a feedback inhibition effect on sulfotransferases was dephosphorylated to generate AMP, which was designed as Module III. Hence, identification of a PAP 3'-phosphatase (CysQ) (EC 3.1.3.7) that specifically hydrolyzes PAP but not PAPS would be of great importance. After expression and purification

(Supplementary Fig. 2a), the enzymatic activities toward PAP and PAPS were determined (Supplementary Fig. 2b). The *Kp*CysQ from *Komagataella phaffii* GS115 that exhibits a high enzymatic activity (81.6 U/mg) (Fig. 1e) was selected and applied for AMP regeneration after confirming its inability towards PAPS (Fig. 1f).

### Characterization of the engineered PAPS regeneration system

After optimizing the committed enzymes for Module I and Module III, the sulfonation efficiency of the PAPS regeneration system was compared with that of the acyclic sulfonation systems (Fig. 2a). As examples, chondroitin 4-*O*-sulfotransferase[18] (C4ST) (Fig. 2b) and trehalose sulfotransferase[39] (Stf0) (PDB ID: 1TEX) (Fig. 2d, Supplementary Figs. 3 and 4a) were introduced for sulfonating chondroitin and trehalose to generate chondroitin sulfate A (CSA)[40] and trehalose-2-sulfate, respectively. After modification, the produced CSA and trehalose-2-sulfate were pretreated and analyzed by liquid chromatography with tandem mass spectrometry (HPLC-MS/MS, Supplementary Fig. 3b, Fig. 4b, c). As shown in Fig. 2c and e, both reactions used to catalyze the transformation of PAP to AMP (Module III) and the regeneration of ATP from AMP (Module I), significantly increased the sulfonation degree and the Active Sulfonate Group Equivalent values (ASGE values, the amount of sulfonate groups transferring to substrate that driven by the initial equal ATP). In comparison, when the complete PAPS regeneration system was applied, the sulfonation degrees of products CSA and trehalose-2-sulfate reached the highest values of 91% and 95%, respectively (Supplementary Figs. 3a and 4d). Correspondingly, the ASGE values of the CSA and trehalose-2-sulfate catalytic systems reached 4.76 and 3.82, respectively.

In parallel, we also compared the current PAPS regeneration system with the commonly used AST IV-dependent recycling system. Obviously, the PAPS regeneration system with $polyP_6$ (energy source) and $SO_4^{2-}$ (sulfonate donor) produced higher sulfonation degree (Fig. 2f, Supplementary Fig. 5). In general, metal ions for instance $Mg^{2+}$, $Ca^{2+}$, and $Mn^{2+}$ were indispensable for enzymatic reactions. Thus, we further investigated the probable effects of these ions on sulfonation reaction. As shown in Supplementary Fig. 6, no significant influence was detected, suggesting the good compatibility of this PAPS regeneration system. Moreover, the purification of the product trehalose-2-sulfate with anion exchange chromatography (Supplementary Fig. 7) also demonstrated the negligible effects of residual polyphosphate. Furthermore, the ready accessibility of these enzymes (Supplementary Table 1) would also facilitate the scaling up of this PAPS regeneration system.

### Re-construction of the PAPS regeneration system for whole-cell transformation

In view of high efficiency of the PAPS regeneration system for sulfonation, it would be also attractive to introduce this regeneration system into living cells. Thus, we firstly investigated the ability of *E. coli* cells for absorbing polyphosphate substrates such as $polyP_6$ with a fluorescein-depending ATP indicating system (Fig. 3a). Briefly, the firefly luciferase[41] (converting luciferin to oxyluciferin) and $PPK2^{s.e}$ (catalyzing AM(D)P to ATP with $polyP_6$ as phosphate donor) encoding genes were introduced into *E. coli* cells. After cultivation with the addition of 6.0 g/L $polyP_6$, *E. coli* cells were harvested and incubated with fluorescein, and then the intracellular oxyluciferin intensity that reflecting the concentration of ATP was detected. As shown in Fig. 3b, stronger fluorescence intensity was observed with addition of $polyP_6$ during cultivation. Meanwhile, ATP analysis results also confirmed that addition of $polyP_6$ during cultivation resulted in much higher level of intracellular ATP (Fig. 3c). The results demonstrated the ability of *E. coli* cells to assimilate extracellular $polyP_6$. Then, all the enzymes of Modules I, II and III were introduced into *E. coli* cells to re-construct an in vivo PAPS regeneration system (Fig. 3d). In order to enhance the uptake of sulfate (Supplementary Fig. 8), the sulfate transporter gene cluster *cysPUWA* in the genome was upregulated with the strong T7

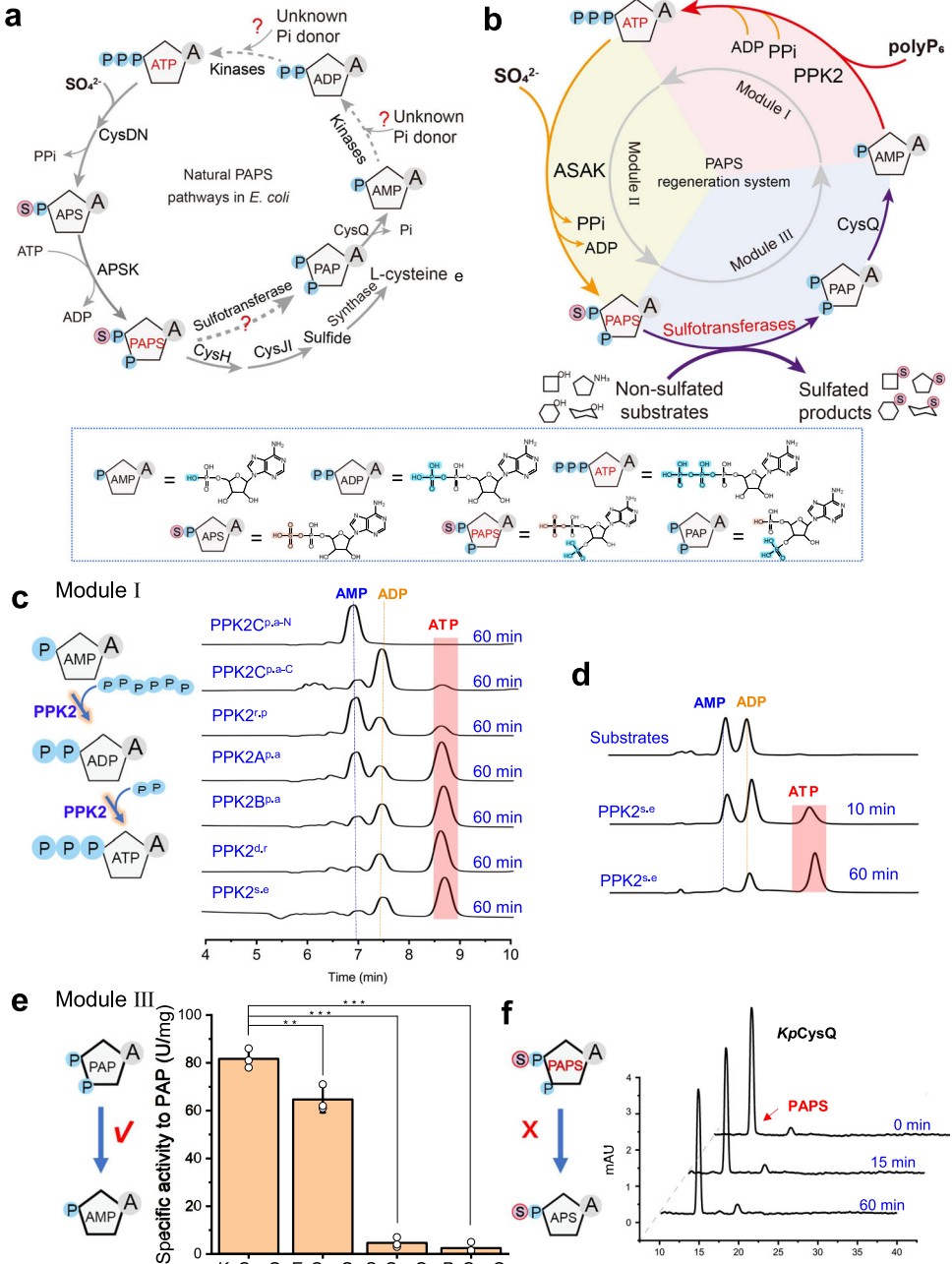

**Fig. 1 | Construction of the PAPS regeneration system. a** Natural PAPS synthesis and decomposition pathways in *E. coli*. **b** The design of the PAPS regeneration pathway. **c** Enzyme activity comparison between different PPK2s using AMP as a substrate. PPK2A^p.a, PPK2B^p.a, PPK2C^p.a, PPK2C^p.a-N (ppk2C^p.a(A1-G753)), and PPK2C^p.a-C (ppk2C^p.a(C748-G1491)) are from *P. aeruginosa*, PPK2^r.p is from *R. pomeroyi*, PPK2^s.e is from *S. epidermidis*, and PPK2^d.r is from *M. ruber*. The catalytic reaction time was 60 min. **d** Comparison of the affinity of PPK2^s.e to AMP and ADP. **e** PAP 3′-dephosphatase enzyme activity assays from different species. *cg*CysQ, *C. glutamicum*. *ec*CysQ, *E. coli* str. K-12 substr. MG1655. *pa*CysQ *P. aeruginosa* PAO1. *sc*CysQ, *S. cerevisiae* S288C.

*kp*CysQ, *K. phaffii* GS115. Significance (*P* value) was evaluated by two-sided *t*-test, *, **, *** denote *P* value < 0.05, <0.01, <0.001, respectively. **f** Conformation that PAP 3′-dephosphatase from *K. phaffii* exhibits no PAPS degradation activity. ATP, adenosine 5′-triphosphate. ADP, adenosine 5′-diphosphate. AMP, adenosine 5′-monophosphate. APS, adenosine-5′-phosphosulfate. PAPS, 3′-phosphoadenosine-5′-phosphosulfate. PAP, 3′-phosphoadenosine-5′-phosphate, polyP₆, hexametaphosphate. PPK2, polyphosphate kinase family 2. ASAK, bifunctional ATP sulfurylase-APS kinase. All the data are expressed as the mean ± S.D. from three (*n* = 3) biologically independent replicates. Source data are provided as a Source Data file.

promoter. As an example, the sulfotransferase AST IV with broad substrate spectrum[42] was overexpressed with N-terminal fusion of GST tag (Supplementary Fig. 9a) in *E. coli* for whole-cell transformation of zosteric acid (a natural anti-adhesive agent, Supplementary Fig. 9bc) from *p*-coumaric acid (Fig. 3d). As shown in Fig. 3e, f, it could be found that introduction of the PAPS regeneration system resulted in a 14.3-fold increase for zosteric acid biosynthesis (1.51 g/L). When further enhancing sulfate uptake, the titer of zosteric acid was increased to

1.73 g/L from 2.0 g/L substrate *p*-coumaric acid, with a conversion rate of 86% (Fig. 3f). The results proved the PAPS regeneration system constructed here could also be applicable for living cells.

## APS is also a universal active sulfonate donor

In nature, NADPH and its unphosphorylated counterpart NADH are important cofactors involved in many oxidation-reduction reactions and can be recognized by specific oxidoreductases[43–45]. Similarly, APS

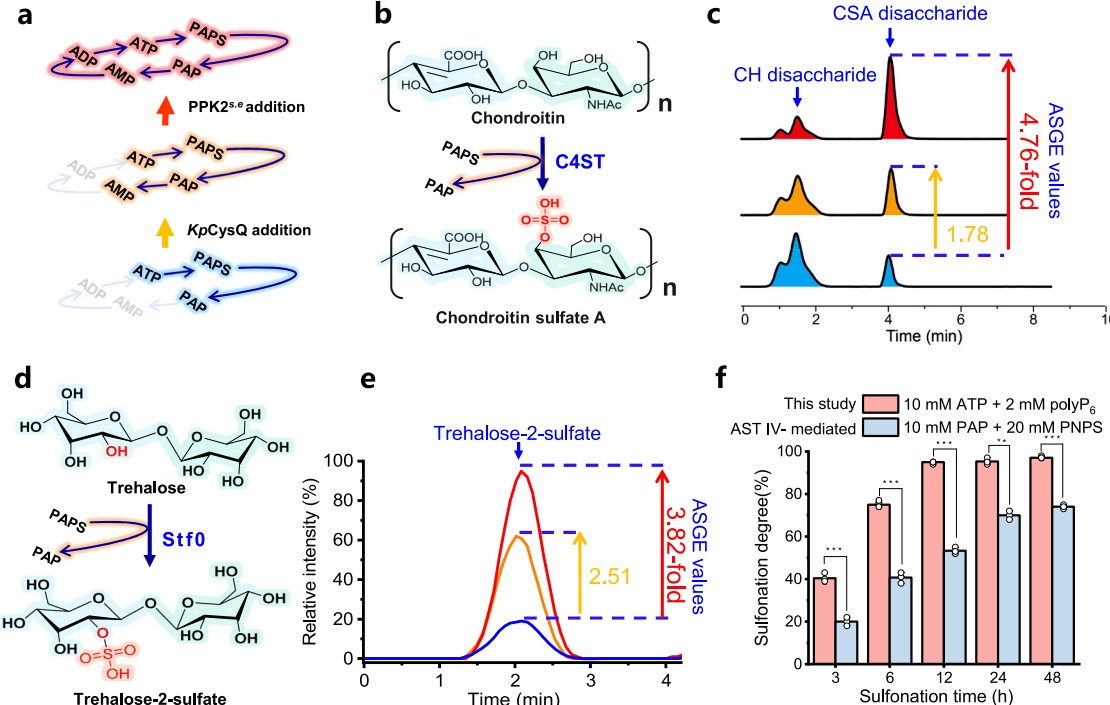

**Fig. 2 | Application and verification of the PAPS regeneration system.**
**a** Schematic diagram showing the step-by-step verification of the PAPS regeneration system. **b** Schematic diagram showing the sulfonation modification of chondroitin sulfate A with the PAPS regeneration system. **c** Quantification of the ASGE values of each step of the PAPS regeneration system in the production of chondroitin sulfate A (CSA). **d** Schematic diagram showing the sulfonation modification of trehalose-2-sulfate with the PAPS regeneration system. **e** Quantification of the ASGE values of each step of the PAPS regeneration system in the production of trehalose-2-sulfate. **f** Sulfonation of trehalose at different times by using the current

PAPS regeneration system and the AST IV-mediated co-factor recycling system. Yellow: with PAP 3'-dephosphatase added, red: with polyphosphokinase 2 added. PPK, polyphosphate kinase. PAPS, 3'-phosphoadenosine-5'-phosphosulfate. PAP, 3'-phosphoadenosine-5'-phosphate. ASGE values, active sulfonate group equivalent values. Significance ($P$ value) was evaluated by two-sided $t$-test, *, **, *** denote $P$ value < 0.05, <0.01, <0.001, respectively. All the data are expressed as the mean ± S.D. from three ($n = 3$) biologically independent replicates. Source data are provided as a Source Data file.

is the unphosphorylated form of PAPS (Fig. 4a). Therefore, we speculated that APS might also be an active sulfonate donor for sulfotransferases. To this end, we selected both cytosolic and membrane sulfotransferases (Fig. 4b), including Stf0 (Fig. 4c), heparan sulfate *N*-sulfotransferase (NST) (Fig. 4d), chondroitin 4-*O*-sulfotransferase (C4ST) (Fig. 4e), estrogen sulfotransferase (Fig. 4f) and AST IV (Fig. 4g), and studied their activities with APS or PAPS as substrates. Interestingly, HPLC-MS/MS (Supplementary Fig. 10) and kinetic parameter (Table 1) analysis results showed that similar to PAPS, APS can also be recognized and utilized by all the above sulfotransferases. The results countered our long-standing acknowledgment that PAPS is the sole active sulfonate donor in living cells.

### Biosynthesis of sulfated compounds with an APS regeneration system

We attempted to construct a short APS generation system for sulfonated modification, in which APS was converted to AMP in one reaction step and only one reproducible ATP was needed as a scaffold (Fig. 5a, Supplementary Fig. 11). Moreover, after transferring the sulfonate group, the product AMP could be directly phosphorylated to ATP by PPK2 in the APS regeneration system. After construction, the ASGE values of Stf0 and C4ST coupling with the APS regeneration system were determined as 2.18 and 2.68, respectively (Supplementary Fig. 12). In view of the broad substrate spectrum[17,46], the sulfotransferase AST IV was chosen for sulfonating a series of benzene ring containing compounds (Fig. 5b) with the APS regeneration system, including zosteric acid, sulfated 4-acetamidophenol (Supplementary Fig. 13a), sulfated estrogen (Supplementary Fig. 13b), sulfated dopamine (Supplementary Fig. 14 and sulfated naringenin (Supplementary

Figs. 15 and 16), which have been widely used as important drugs and building blocks[47–49]. Overall, the results suggest that the engineered APS regeneration system shows great potential for cell-free and in vivo sulfonation reactions.

### Development of an APS-mediated sulfotransferase activity indicating system

Compared with the PAPS regeneration system, the APS regeneration system achieved comparatively lower ASGE values towards sulfotransferases (Fig. 2, Supplementary Fig. 12). Hence, to construct and rapidly characterize sulfotransferase variants with higher activities towards APS, an APS-mediated sulfotransferases activity indicating system was proposed and established. In the APS regeneration system, PPi, AMP, and ATP as intermediate products were produced (Fig. 5a) while it has been reported that pyruvate phosphate dikinase (*mm*PPDK) from *Methanosarcina mazei*[50] catalyzes the interconversion of AMP, PPi, and phosphoenolpyruvate (PEP) with ATP, Pi and pyruvate (Fig. 6a). As a consequence, we recruited the specific sulfotransferase Stf0, *mm*PPDK and the lactate dehydrogenase (*pa*LDHA for the conversion of pyruvate and NADH to lactic acid and NAD⁺) to accomplish the rapid analysis of sulfonation that driven by APS with detection of the absorbance decrease at 340 nm (Fig. 6b). Specifically, all the enzymes were expressed with *E. coli* cells and purified for next step verifying (Supplementary Fig. 17). After successfully engineering the indicating system, the effects of all the components of the system on the reaction that catalyzed by *pa*LDHA were studied. The results showed that ATP, ADP, AMP, PEP, and PPi generated no obvious negative effects while Mg²⁺ produced a positive effect (Supplementary Fig. 18). Further examination of the absorbance-drop values at 340 nm

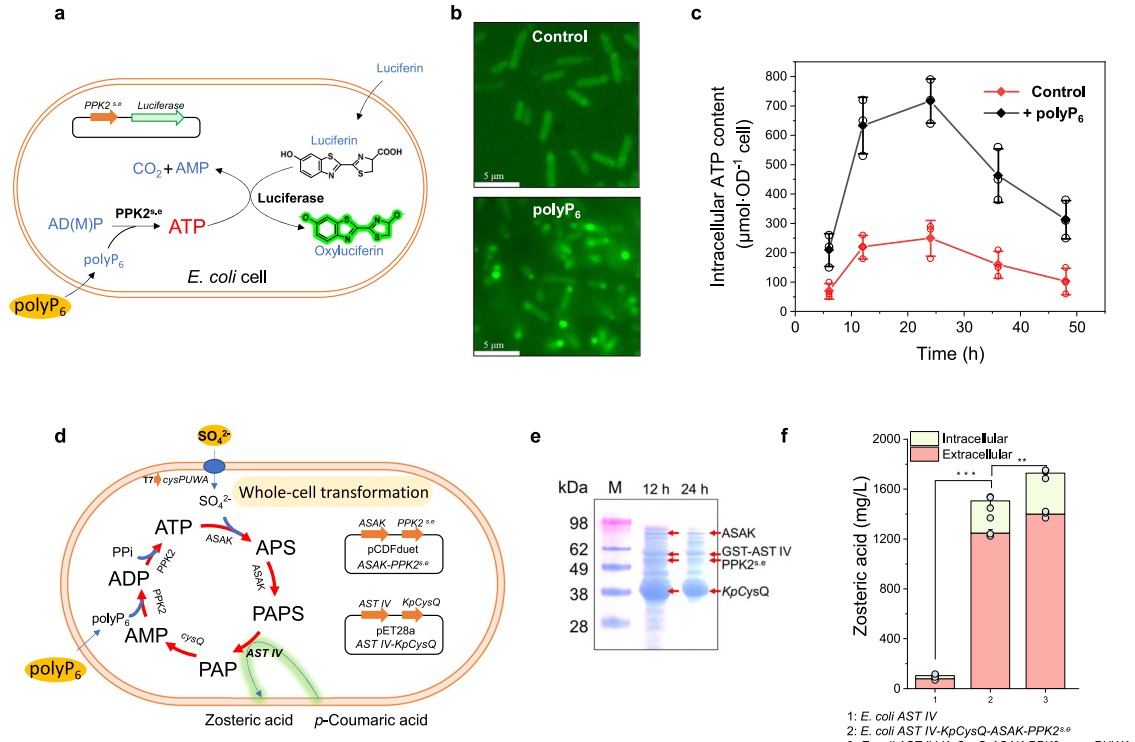

**Fig. 3 | Re-construction of an in vivo PAPS regeneration system for whole-cell transformation. a** Characterization of intracellular ATP content schematic diagram. **b** Microscopic observation of cell morphology and brightness, the higher the brightness of the cell indicates the higher the ATP content in the cell. Scale bar, 5 µm. Three independent experiments were performed, with similar results. **c** Impact of polyP$_6$ supplementation on the intracellular ATP concentrations. **d** Schematic diagram of an in vivo PAPS regeneration system for whole-cell transformation. **e** SDS-PAGE analysis of protein expression in engineered *E. coli*. Three independent experiments were performed, with similar results. **f** Determination of the yield of zosteric acid produced by the engineered bacteria. Significance (*P* value) was evaluated by two-sided *t*-test, *, **, *** denote *P* value < 0.05, <0.01, <0.001, respectively. ATP, adenosine 5′-triphosphate. ADP, adenosine 5′-diphosphate. AMP, adenosine 5′-monophosphate. APS, adenosine-5′-phosphosulfate. PAPS, 3′-phosphoadenosine-5′-phosphosulfate. PAP, 3′-phosphoadenosine-5′-phosphate. All the data are expressed as the mean ± S.D. from three (*n* = 3) biologically independent replicates. Source data are provided as a Source Data file.

and the consistent liquid chromatography-mass spectrometry assay results demonstrated the reliability of this indicating system (Fig. 6c).

## Semi-rational engineering of sulfotransferases to enhance APS utilization efficiency

After engineering the above rapid indicating system, we concentrated on uncovering the key residues affecting substrate recognition and enhancing APS utilization efficiency. Accordingly, two conserved 5′PSB and 3′PB motifs that interact with the 5′-phosphosulfate group and 3′-phosphate group of PAPS, respectively, were displayed in the local 3D structure of Stf0 sulfotransferase[19,39,51] (PDB: 1TEX) (Fig. 7a; Supplementary Fig. 19). According to previous studies, the residues arginine and serine of the 3′PB motif should help position PAPS in sulfotransferases[52,53]. Thus, the residues R143, V151, S152, and W154 in Stf0, and R132, V139, S140, and Y141 in AST IV were selected for saturation mutation after further multiple sequence alignment (Fig. 7b). The mutation process was followed as Fig. 7c and the enzyme activity was measured according to the APS-mediated indicating system (Fig. 6b). As shown in Fig. 7d, the single mutations R143C, S152P, and W154N dramatically improved the activity of Stf0 for APS. The variant Stf0$^{R143C/W154N}$ with the highest activity towards APS (comparable to PAPS) was further constructed by combinatorial mutation (Fig. 7f). For AST IV, the positive single mutations are R132N, V139G, S140I, and S140E (Fig. 7e) while the best combinatorial variant is AST IV$^{R132E/S139G/S140I}$ (Fig. 7g). These results suggest that reprogramming of the 3′PB motif of sulfotransferases can improve the activity and affinity (Supplementary Table 2) towards APS. Eventually, the constructed combinatorial variants Stf0$^{R143C/W154N}$ and AST IV$^{R132E/S139G/S140I}$ were used for sulfonating trehalose and *p*-coumaric acid with the APS regeneration system.

Compared with the wild type Stf0 and AST IV (Fig. 5b, Supplementary Fig. 12), Stf0$^{R143C/W154N}$ and AST IV$^{R132E/S139G/S140I}$ with higher affinity to APS generated significantly increased ASGE values, which were 3.26 (85% yield) and 4.03 (89% yield), respectively (Fig. 7h, i). These results demonstrated that sulfonated modification with efficient APS-mediated sulfonate group regeneration systems by rational engineering of sulfotransferases is feasible and convenient for practical applications.

## Discussion

Performing enzymatic modification through sulfonation involving the consumption of PAPS and accumulation of the byproduct PAP is always a challenging task. In this study, we designed and constructed a de novo PAPS regeneration system by assembling the following three modules: the biosynthesis of ATP from AMP (Module I), the conversion of PAPS from ATP (Module II) and the regeneration of AMP from PAPS (Module III) (Fig. 1b). As a result, we achieved the enzymatic biosynthesis of sulfated compounds including chondroitin sulfate A (Fig. 2d) and trehalose-2-sulfate (Fig. 2e), by providing inexpensive substrate sulfate (SO$_4^{2-}$ as sulfonate donor) and polyP$_6$ (as energy source). Compared to the traditional PNPS-dependent system used for PAPS regeneration[20], this route is more promising since no high-cost substrates were added and no toxic byproducts such as PNP, accumulated during sulfonation and PAPS regeneration (Fig. 1b). Moreover, compared with the biotransformation system by directly providing PAPS[22,54,55], this PAPS regeneration system produced higher sulfonation degree (Fig. 2, Supplementary Fig. 3, Table 3), the reason should be ascribed to the continuous supply of PAPS and the elimination of PAP accumulation. It has been reported that the presence of PAP, especially with a high concentration[55], should affect PAPS utilization

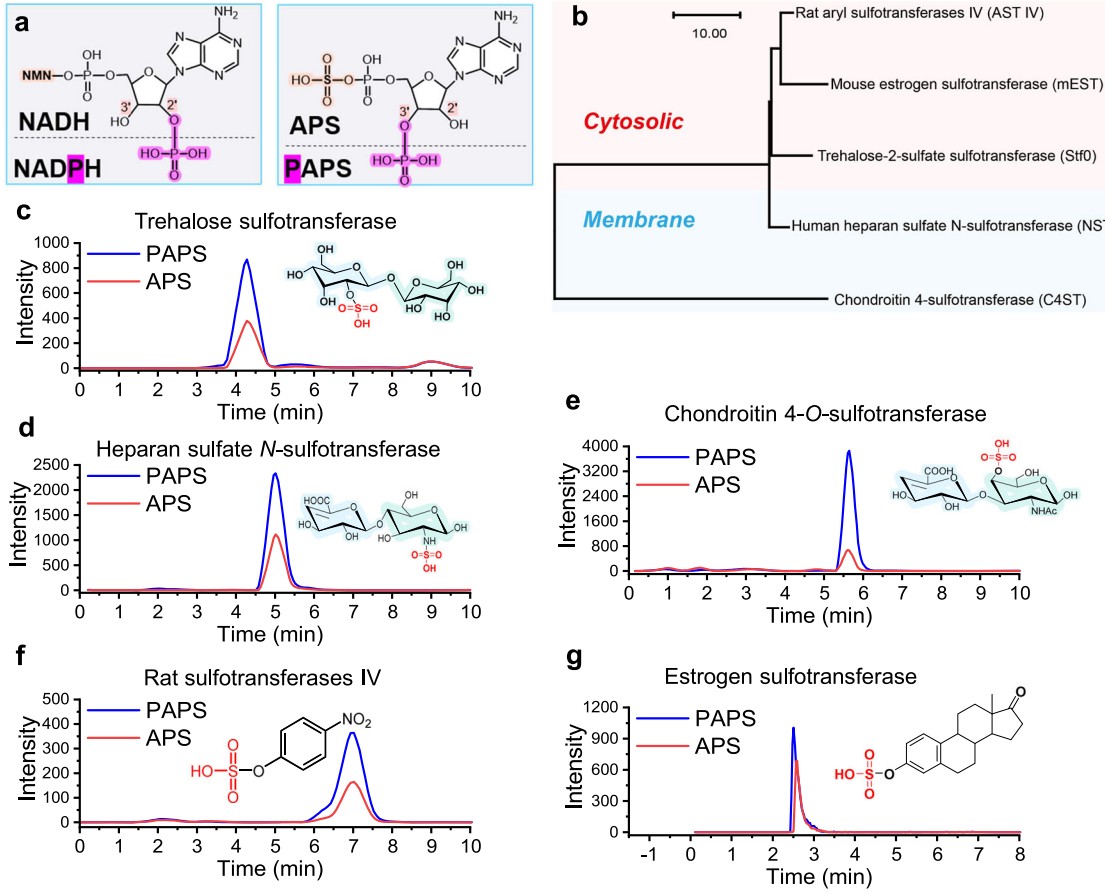

**Fig. 4 | Verification of APS as an active sulfonate group donor for biological sulfonation reactions. a** Structure comparison of NADPH and NADH, PAPS, and APS. **b** Phylogenetic analysis of the sulfotransferase family. **c–g** Verification and comparison of sulfonate transferase enzyme activity by using PAPS or APS as sulfonate donor. **c** trehalose sulfotransferase using APS as the sulfonate donor; **d** heparan sulfate *N*-sulfotransferase using APS as the sulfonate donor, **e** chondroitin 4-*O*-sulfotransferase using APS as the sulfonate donor; **f** rat sulfotransferase IV using APS as the sulfonate donor; **g** estrogen sulfotransferase using APS as the sulfonate donor. APS adenosine-5'-phosphosulfate. PAPS 3'-phosphoadenosine-5'-phosphosulfate. NADPH nicotinamide adenine dinucleotide phosphate. NADH nicotinamide adenine dinucleotide. NMN nicotinamide mononucleotide. All the data are expressed as the mean ± S.D. from three (*n* = 3) biologically independent replicates. Source data are provided as a Source Data file.

since the continued occupation of PAP in the sulfotransferase pocket resulted in the prevention of PAPS binding[34]. In addition, after demonstrating the ability of *E. coli* cells for absorbing extracellular polyP$_6$ and successfully re-constructing the in vivo PAPS regeneration system (Fig. 3), it could be concluded that this constructed PAPS regeneration system should have broad applications in the fields of multi-enzyme catalysis and metabolic engineering especially for sulfated products[56].

Generally, the ATP regeneration system (Module I) requires different classes of polyphosphokinases for cascade phosphorylation of AMP and ADP[30]. Here, considering the AMP and ADP phosphorylase activities of PPK2 observed[57], we comparatively investigated PPK2 from different species and finally selected PPK2$^{s.e}$ to perform the direct biocatalytic synthesis of ATP from AMP (Fig. 1c). The application of PPK2 containing both AMP phosphorylase and ADP phosphorylase improves regeneration efficiency and simplifies ATP regeneration in various scenarios. Furthermore, our results also demonstrated that PPK2$^{s.e}$ exhibits higher affinity toward AMP and preferentially phosphorylates AMP to ATP through ADP (Fig. 1c, d). These results increase our knowledge on the enzymatic properties of PPK2 family enzymes[58]. In parallel, the identification of *Kp*CysQ from *K. phaffii* GS115 for selectively breaking down PAP to AMP (Fig. 1e) but not PAPS to APS (Fig. 1f) suggests that CysQ should have an identification mechanism to distinguish PAP and PAPS (Fig. 1f) even though it has been reported that CysQ as part of the

larger FIG superfamily of phosphatases exhibits dephosphorylation activities toward different monosaccharide-containing substrates[59,60].

To date, studies on sulfur metabolism have mainly concentrated on pathogenic and sulfur bacteria[61–63]. It has always been accepted that PAPS is the sole universal sulfate donor in all biochemical

**Table 1 | Kinetic parameters of sulfotransferases**

| Enzyme | Substrate | $K_m$ (mM) | $k_{cat}$ (h$^{-1}$) | $k_{cat}/K_m$ (h$^{-1}$ mM$^{-1}$) |
|---|---|---|---|---|
| StfO | APS | 1.56 ± 0.26 | 21.24 ± 1.44 | 13.62 |
| | PAPS | 0.82 ± 0.06 | 35.58 ± 0.90 | 43.39 |
| NST | APS | 2.72 ± 0.41 | 47.02 ± 1.43 | 17.29 |
| | PAPS | 1.14 ± 0.15 | 72.50 ± 0.91 | 63.60 |
| C4ST | APS | 2.65 ± 0.52 | 21.86 ± 2.15 | 8.25 |
| | PAPS | 1.27 ± 0.14 | 55.02 ± 2.30 | 43.32 |
| AST IV | APS | 2.45 ± 0.50 | 24.78 ± 2.46 | 10.11 |
| | PAPS | 1.43 ± 0.172 | 47.3 ± 2.21 | 33.08 |
| EST | APS | 2.91 ± 0.52 | 25.56 ± 2.41 | 8.78 |
| | PAPS | 1.56 ± 0.26 | 30.58 ± 2.06 | 19.60 |

All the data are expressed as the mean ± SD from three (*n* = 3) biologically independent replicates. Source data are provided as a Source Data file.

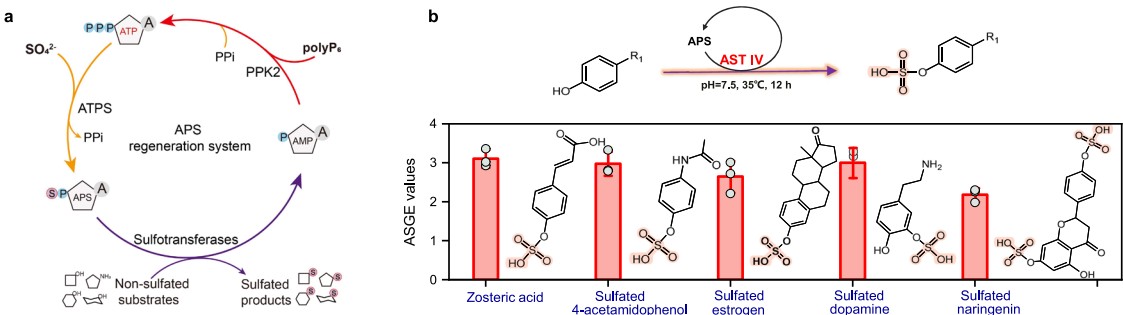

**Fig. 5 | Construction of a refined sulfonate regeneration system mediated by APS. a** Design of the APS-sulfonate donor regeneration system. **b** Compound sulfonation modification catalyzed by AST IV using the APS regeneration system. ATP adenosine 5'-triphosphate, ADP adenosine 5'-diphosphate, AMP adenosine 5'-monophosphate, APS adenosine-5'-phosphosulfate, PAPS, 3'-phosphoadenosine-5'-

phosphosulfate, PolyP$_6$ hexametaphosphate, PPK polyphosphate kinase, PPi, pyrophosphate, ATPS, ATP sulfurylase. ASGE values, active sulfonate group equivalent values. All the data are expressed as the mean ± S.D. from three ($n$ = 3) biologically independent replicates. Source data are provided as a Source Data file.

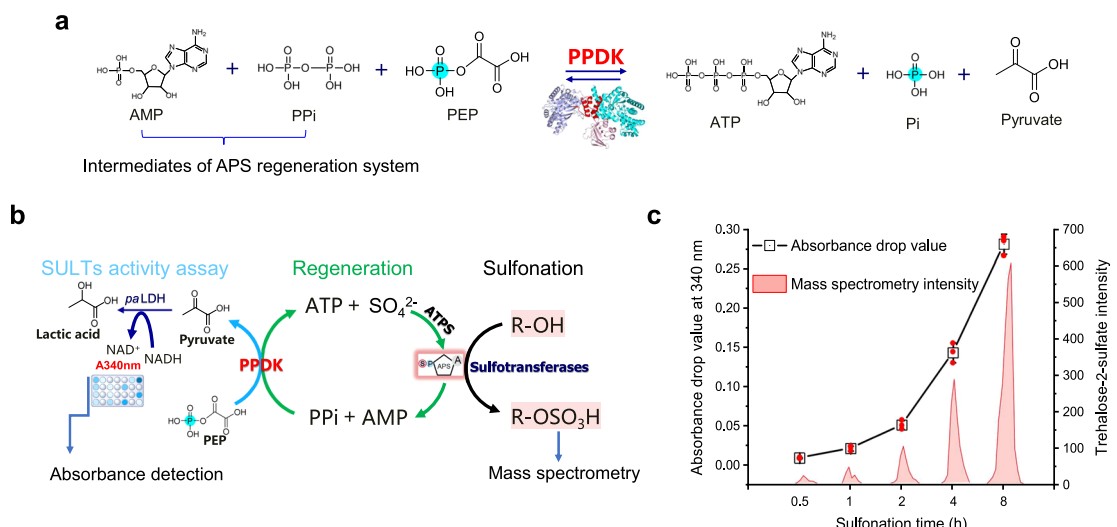

**Fig. 6 | Construction of an APS-mediated sulfotransferase activity indicating system. a** The reaction equation catalyzed by PPDK. **b** Schematic diagram of the APS-mediated sulfotransferase activity indicating system. **c** Absorbance detection and LC-MS detection were used to verify the APS-mediated sulfotransferases activity indicating system by using Stf0 as an example. APS adenosine-5'-phosphosulfate, PPDK pyruvate phosphate dikinase, LDHA lactate dehydrogenase A,

PEP phosphoenolpyruvate, AMP adenosine 5'-monophosphate, PPi pyrophosphate, ATP adenosine 5'-triphosphate, NADH (NAD$^+$), reduced (oxidized) nicotinamide adenine dinucleotide. All the data are expressed as the mean ± S.D. from three ($n$ = 3) biologically independent replicates. Source data are provided as a Source Data file.

reactions. Here, we demonstrated that in addition to PAPS, APS is also an active sulfate donor and can be recognized by different cytosolic and membrane sulfotransferases (Fig. 4b). This finding provides insights into our understanding of the physiological functions of APS and PAPS. Moreover, from the perspective of application, we simplified the PAPS regeneration system (Fig. 1b) into an APS regeneration system (Fig. 5a), resulting in the continuous conversion of APS from sulfate and polyP$_6$ with one molecule of ATP as the intermediate (Fig. 5a). Moreover, after construction of an APS-mediated sulfotransferase activity indicating system by adopting PPDK[50], LDHA and specific sulfotransferase (Fig. 6b), we semi-rationally modified sulfotransferases Stf0 and AST IV and rapidly screened the variants of Stf0$^{R143C/W154N}$ and AST IV $^{R132E/S139G/S140I}$ with higher APS activities (Fig. 7d, e). As a consequence, specific efficient APS regeneration systems were developed for sulfonating trehalose (Fig. 7h) and p-coumaric acid (Fig. 7i), respectively. The results suggest that with the help of the APS-mediated sulfotransferase activity indicating system, it would be convenient and effective to engineer sulfotransferases for developing APS regeneration systems with high capacity.

Overall, a PAPS regeneration system from sulfate and polyphosphate was designed and successfully constructed in this study by assembling three modules. After demonstrating the ability of *E. coli* cells for absorbing polyP$_6$, the PAPS regeneration system was introduced into living cells for whole-cell transformation. Then, after identifying APS as another active sulfonate group donor, a more simplified APS regeneration system was further engineered for sulfonated modification. In particular, an APS-mediated sulfotransferase activity indicating system was established for rapidly screening sulfotransferase variants with satisfactory activity and affinity to APS. The alternative PAPS and APS regeneration systems for sulfonated modification were environment-friendly and applicable to be scaled-up for boosting the biomanufacturing of various sulfated natural products.

## Methods

### Bacterial strains and reagents

*Escherichia coli* strain JM109 was used for all cloning experiments, and *E. coli* strain BL21(DE3) was used for protein expression. Adenosine 3'-phosphoadenosine 5'-phosphosulfate triethylammonium salt (>90%), adenosine 5'-phosphosulfate sodium salt (>85%), sodium

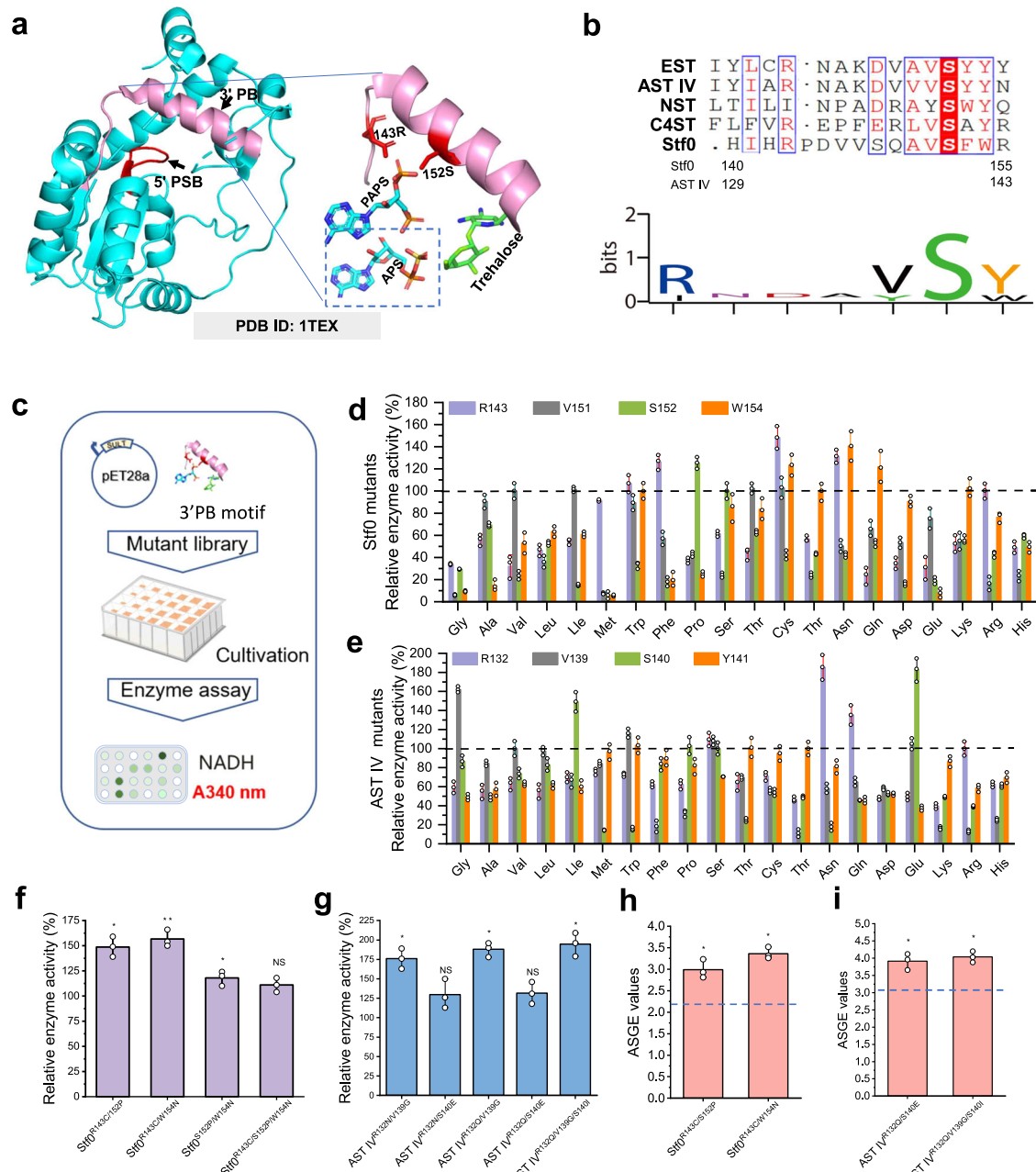

**Fig. 7 | Modifying the 3'PB motif to enhance APS utilization efficiency of sulfotransferases. a** Localization of the 5'PSB and 3'PB motifs in the trehalose sulfotransferase structure. Red indicates 5'PSB. Pink indicates 3'PB. Left: overall structure of Stf0, right: docking of PAPS and APS to trehalose sulfotransferase's structure. **b** Sequence analysis of 3'PB motifs from different sulfotransferases. Upper: multiple sequence alignment, lower: high-frequency analysis of amino acids in 3'PB motifs. **c** Mutants' construction and enzyme activity verification process. **d** Enzyme activity assay of saturation Stf0 mutation. **e** Enzyme activity assay of saturation AST IV mutation. **f** Enzyme activity assay of superposition mutation of Stf0. **g** Enzyme activity assay of superposition mutation of AST IV. **h** Quantification

of the ASGE values of different Stf0 mutants coupled with the APS regeneration system. **i** Quantification of the ASGE values of different AST IV mutants coupled with the APS regeneration system, dotted lines indicate the ASGE values of WT enzymes. APS adenosine-5'-phosphosulfate, PAPS 3'-phosphoadenosine-5'-phosphosulfate, 5'PSB 5'-phosphosulfate group, 3'PB, 3'-phosphate group, NADH reduced nicotinamide adenine dinucleotide. ASGE values, active sulfonate group equivalent values. Significance (*P* value) was evaluated by two-sided *t*-test, *, **, *** denote *P* value < 0.05, <0.01, <0.001, respectively. NS, not significant (*P* > 0.05). All the data are expressed as the mean ± S.D. from three (*n* = 3) biologically independent replicates. Source data are provided as a Source Data file.

pyrophosphate (>99%), and sodium hexametaphosphate (>96%) were purchased from Sigma-Aldrich (Shanghai, China). Common reagents were purchased from Sinopharm (Beijing, China).

### Genetic construction
All the primers designated and plasmids constructed in this work are listed in Supplementary Data 1, 2, and 3. Artificial bifunctional PAPS synthase (ASAK) contains ATP sulfurylase from *Saccharomyces*

*cerevisiae* and APS kinase from *E. coli* K12[32]. The polyP kinases genes ppk2A[pa] (Gene ID: 879494), ppk2B[pa] (Gene ID: 882843), ppk2C[pa] (Gene ID: 878853, ppk2C[paN] (ppk2C[pa(a1-g753)]), ppk2C[paC], ppk2C[pa(c748-g1491)]), ppk2[rp] (UniProtKB: Q5LX16, *Ruegeria pomeroyi*), ppk2[se] (UniProtKB: A0A6N3RU13, *S. epidermidis*), and ppk2[dr] (UniProtKB: M9XB82, *Meiothermus ruber*) were amplified or synthesized and inserted into the pET28a(+) plasmid[32]. Codon-optimized genes of trehalose sulfotransferase (Stf0, Sequence ID: AWT51695.1), estrogen

sulfotransferase (EST, Sequence ID: P49891.2), firefly luciferase (luciferase, GenBank: BAM38523.1) and pyruvate phosphate dikinase (*mm*PPDK, Sequence ID: 44088594) were synthesized by Sangon Biotech (Shanghai, China) and inserted into the pET28a (+) plasmid. 3′ (2′),5′-bisphosphate nucleotidase *cg*CysQ (GeneID: 58311165, *Corynebacterium glutamicum*, https://www.ncbi.nlm.nih.gov/protein/WP_003858358.1/), *ec*CysQ (*E. coli* str. K-12 substr. MG1655, GeneID: 948728), *pa*CysQ (*P. aeruginosa* PAO1, GeneID: 881784), *sc*CysQ (*S. cerevisiae* S288C, GeneID: 854090), *kp*CysQ (GeneID: 8198929, *K. phaffii* GS115), and *pa*LDHA (GeneID: 881895, *P. aeruginosa* PAO1) were amplified and inserted into the pET28a (+) plasmid using the T5 exonuclease DNA assembly (TDEA) method[64].

### Genomic promoter replacement
Genome integration and promoter replacement was performed using the CRISPR-Cpf1 gene editing system, and the editing process was mediated by the pEcCpf1/pcrEG double-plasmid system[65]. Two homologous sequences of the template were amplified from the genome of *E. coli* BL21(DE3) using the primer pairs CysP3-F/CysP3-R and CysP4-F/CysP4-R. Primer pairs CysP1-F/CysP1-R, CysP2-F/CysP2-R were used for the construction of the sgRNA expression cassette using the plasmid pcrEG as a template. All four fragments were assembled to generate the plasmid pcrEG-*CysPUWA* using the TDEA method.

### Protein expression and purification
After an overnight incubation in LB medium at 37 °C, recombinant *E. coli* strains were transferred to TB medium and cultivated for 8−12 h at 30 °C, and 220 rpm, and 0.5 mmol/L isopropyl-β-D-thiogalactopyranoside was added to induce target protein expression. Cells were collected by centrifugation (5000 × *g*, 4 °C, 15 min) and disrupted by sonication (50 mM Tris-HCl buffer pH 7.5, ice bath). The supernatant was obtained by centrifugation (10,000 × *g*, 4 °C, 30 min).

Proteins were purified from the supernatant of the cell lysates by nickel-affinity chromatography with AKTA start 25 (GE Healthcare, USA) with washing buffer (50 mM Tris-HCl, pH 7.5, 150 mM NaCl) and elution buffer (50 mM Tris-HCl, pH 7.5, 500 mM imidazole, 150 mM NaCl). The purified protein solutions were desalted using a HiTrap desalting column (GE Healthcare, USA) with washing buffer (50 mM Tris-HCl, pH 7.5, 150 mM NaCl). Protein concentration was determined using a Bradford protein assay kit from Beyotime (Shanghai, China) using bovine serum albumin as the standard. Sodium dodecyl sulfate-polyacrylamide gel electrophoresis (PAGE) analysis was performed using 10−12% bis-tris protein gels (NuPAGE, Thermo Scientific), and the gels were stained with Coomassie Brilliant Blue solutions.

### Activity assay of the polyP kinases
The phosphorylation reaction was carried out in 50 mM Tris-HCl, pH 7.5, 150 mM NaCl, buffer, containing 10 mM ADP (or 10 mM AMP or both 5 mM ADP and 5 mM AMP), 5.0 g/L polyP$_6$ (polyP$_3$, or PPi), 20 mM MgSO$_4$, and 0.5 g/L purified polyP kinases. After the mixture was incubated at 35 °C for a designated time (0−60 min), the reaction was terminated by adding an equal volume of methanol. One unit of activity was defined as the amount of enzyme needed to catalyze the formation of 1 μmol ATP (or 1 μmol ADP) at 35 °C within 1 min. Specific activity was expressed in unit/mg protein.

### Quantification of AMP, ADP, ATP, PAP, PAPS, and APS
High performance liquid chromatography equipped with an Agilent 1260 system was used for quantification of AMP, ADP, ATP, PAP, PAPS and APS. A YMC-Pack Polyamine II column was used for quantification of PAP, PAPS and APS. HPLC was performed as follows: the composition of the mobile phase was 50 mM KH$_2$PO$_4$ with 0.1% (v/v) triethylamine and 10% acetonitrile. The injected volume of all samples was 5 μL, and the column temperature was set to 30 °C with a flow rate of

0.6 mL/min. Fractions were detected by UV absorbance at a wavelength of 254 nm.

A Hypersil GOLD™ aQ (250 × 4.6 mm, 5 μm, Thermo Scientific™) was used for quantification of AMP, ADP, and ATP. The HPLC detection conditions were as follows: the composition of the mobile phase was 50 mM KH$_2$PO$_4$-K$_2$HPO$_4$, pH 6.7. The injected volume of all samples was 5 μL, and the column temperature was set to 30 °C with a flow rate of 0.8 mL/min.

### Activity assay of sulfotransferases
A reaction mixture containing 50 mM Tris-HCl, pH 7.5, 150 mM NaCl, 5-10 mM PAPS (5−10 mM APS), 10 mM MgSO$_4$, 0.5 mg/mL sulfotransferases and 10 mM (polymers for 5 g/L) nonsulfonated substrate was prepared in a total volume of 1000 μL. The reaction mixture was incubated at 35 °C for 6 h and quenched by adding 1000 μL of methanol. The resultant mixtures were centrifuged at 10,000 × *g* for 10 min. One unit of activity was defined as the amount of enzyme needed to catalyze the formation of 1 mM sulfated products at 35 °C within one hour. The activities of Stf0 sulfotransferase mutants generated in the first-round screening were measured using APS and trehalose as substrates. Crude enzyme was obtained by lysis of five OD bacteria in Tris-HCl buffer.

### Sulfonated products quantification
Sulfonated products were quantified by high performance liquid chromatography-mass spectrometry (Shimadzu LCMS-IT-TOF). Samples were analyzed by an XSelect CSH C18 column, 130 Å, 1.7 μm, 2.1 × 50 mm. The LC conditions were as follows: eluent A (0.1 mol/L ammonium formate) and eluent B (acetonitrile) were used at a flow rate of 0.6 mL/min. MS analyses were simultaneously performed using electrospray ionization in the positive ionization mode (detector voltage 30 V, capillary voltage 3.0 kV, and source block temperature 100 °C) to obtain spectra across a scan range of 20−2000 m/z. To quantify the sulfated products, we isolated the mass spectrum corresponding to the m/z values of the designated products from the parent ion fragments. Subsequently, the concentrations of sulfated products were calculated based on the area of the selected peak. Data were acquired using MassLynx software (version 4.1, ScN870; Waters).

### Preparation of PAPS regeneration and APS regeneration system
The PAPS regeneration system was prepared with a mixture containing 50 mM Tris-HCl buffer (pH 7.0), 10 mM ATP, 10 mM MgSO$_4$, 0−50 mM Na$_2$SO$_4$, 0−5 mM polyP$_6$, 0.5 mg/mL ASAK, 0.5 mg/mL *Kp*CysQ and 0.5 mg/mL PPK2$^{s.e.}$. Sulfotransferases and precursors were subsequently combined. The APS regeneration system was mixed with 50 mM Tris-HCl buffer (pH 7.0), 10 mM ATP, 10 mM MgSO$_4$, 0−50 mM Na$_2$SO$_4$, 0−5 mM polyP$_6$, 0.5 mg/mL ATP sulfurylase from *S. cerevisiae* (ATPS$^s$) and 0.5 mg/mL PPK2$^{s.e.}$.

### Intracellular ATP analysis
Cell samples were collected and washed with sterile saline solution. Fluorescein (5.0 g/L) was added to an equal cell density suspension and then incubated for 10 min at 30 °C. To analyze the cell morphology, an Eclipse Ni-E microscope (Nikon, Tokyo, Japan) was used to observe the cell shape and fluorescence under the phase-contrast lens. To analyze intracellular ATP content, the cells were broken and determined using an ATP content assay kit purchased from Sangon Biotech (Shanghai, China)[32].

### Activity assay of *pa*LDHA and *mm*PPDK
To assay *pa*LDHA activity, a reaction mixture containing 50 mM Tris-HCl, pH 7.5, 4.0 g/L NADH, 5 g/L pyruvate, and 0.5 mg/mL *pa*LDHA was measured by an Infinite 200 PRO plate reader at an absorbance of 340 nm. One unit of activity was defined as the amount of enzyme needed to oxidize 1 μmol NADH at 30 °C within 1 s. To assay *mm*PPDK activity, a reaction mixture containing 50 mM Tris-HCl, pH 7.5, 10 mM

PEP, 10 mM AMP, 10 mM PPi, and 0.5 mg/mL *mm*PPDK was prepared in a total volume of 1000 μL. The reaction mixture was incubated at 35 °C for 0–12 h, and then the production of pyruvate was measured. Pyruvate content was measured by a reaction catalyzed by *pa*LDHA, which caused a decrease in absorbance at 340 nm using NADH as the other substrate. One unit of activity was defined as the amount of enzyme needed to catalyze the formation of 1 μmol pyruvate at 35 °C within 1 h. Specific activity was expressed in unit/mg protein.

### Phylogenetic analysis of sulfotransferases

Sequence alignment was performed using ESPript[66]. A phylogenetic tree of sulfotransferases was constructed with the maximum likelihood method using MEGA-X[67].

### Molecular docking

Molecular docking of *Mycobacterium smegmatis* Stf0 sulfotransferase (PDB ID: 1TEX) with PAPS and APS was performed using Discovery Studio 4.0. The structure of Stf0 for molecular docking was prepared by eliminating all bound water molecules, and adding hydrogen atoms, which was optimized by applying the CHARMM force field. Amino acids in or near the active pocket were selected to create protein conformations and refine side chains[68].

### Mutagenesis

Variants were generated using a whole-plasmid mutagenesis protocol. Saturation mutagenesis was then performed on beneficial sites using the NNK codon[69]. The final mutants were confirmed by DNA sequencing (Sangon Biotech). To assay mutants' activity, a reaction mixture containing 50 mM Tris-HCl, pH 7.5, 20 mM PEP, 10 mM AMP, 10 mM PPi, 0.5 mg/mL PPDK, 0.5 mg/mL sulfotransferases, 0.5 mg/mL ATPS[S], and 10 mM nonsulfonated substrate was prepared in a total volume of 1000 μL. The reaction mixture was incubated at 35 °C for 0-12 h, and then the production of pyruvate by a reaction with changes in absorbance catalyzed by *pa*LDHA was measured.

### Reporting summary

Further information on research design is available in the Nature Portfolio Reporting Summary linked to this article.

## Data availability

Data supporting the findings of this work are available within the paper and its Supplementary Information files. A reporting summary for this Article is available as a Supplementary Information file. Source data are provided with this paper, which are also available at figshare [https://doi.org/10.6084/m9.figshare.24449563][70]. Source data are provided with this paper.

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

## Acknowledgements
This work was financially supported by the National Natural Science Foundation of China (32370066), the National Key Research and Development Program of China (2021YFC2100800), National First-class Discipline Program of Light Industry Technology and Engineering (QGJC20230202), and the Postgraduate Research & Practice Innovation Program of Jiangsu Province (KYCX23_2487).

## Author contributions
Z.K. and R.X. designed the experiments. R.X. performed the experiments. R.X. and Z.K. analyzed the results. R.X. and Z.K. prepared and revised the manuscript. J.C. performed experiments for *E. coli* genome integration. W.Z., X.X., Y.W., G.D., J.L. and J.C. participated in revising the manuscript. Z.K. conceived and directed the project. All authors have given approval to the final version of the manuscript.

## Competing interests
The authors declare no competing interests.
