## [Peer Review File · Nature Communications]

Engineering sulfonate group donor regeneration systems to boost biosynthesis of sulfated compoundsReviewers' Comments:

Reviewer #1:

Remarks to the Author:

Xu et al. introduce a novel 3'-phosphoadenosine-5'-phosphosulfate (PAPS) regeneration system that encompasses modules responsible for converting AMP to ATP, ATP to PAPS, and PAPS to AMP using sulfotransferase. This PAPS and APS biosynthesis system was employed to produce a range of sulfated compounds. Although I found the study intriguing, it seems too preliminary for publication at this stage. Firstly, while the in vitro production of sulfated products using PAPS and APS regeneration systems appeared impressive, the authors need to demonstrate the superiority of their system within cellular biosynthesis (e.g., by expressing all necessary enzymes and examining the production yield of biosynthesized sulfated products). Additionally, several PAPS regeneration systems have been reported in previous studies (e.g., *Biotechnol. J.* 2019, 14, e1800436; *J. Bacteriol.* 1992, 174, 415). The authors should compare the efficiencies of these systems with their engineered system and clearly outline the advantages of their approach over existing systems. Finally, the authors semi-rationally engineered Stf0 to enhance its APS affinity. Why was Stf0 chosen as a target? Did the authors attempt to work with any of the other sulfotransferases mentioned in this study?

I also have a few minor comments:

1. Please include chemical structures in Figure 1.
2. In Figure 1A, compare the engineered system with natural systems.
3. Could the authors indicate the reaction time length in Figure 1B?
4. On line 109, correct the formatting error with "(Fig. 1F" by removing the bold formatting from the opening parenthesis.

Reviewer #2:

Remarks to the Author:

To construct in vitro biochemical sulfonation system, the authors developed phosphoadenosine-5'-phosphosulfate (PAPS, sulfonate group donor) regeneration systems composed of three modules; module 1, ATP regeneration from AMP using polyphosphate kinase; module 2, PAPS synthesis using PAPS synthetase; module 3, transferring sulfonate group from PAPS to target compounds using sulfotransferase and dephosphorylation of phosphoadenosine-5'-phosphate (PAP) to AMP. They demonstrated chondroitin sulfate and trehalose sulfate synthesis using the PAPS regeneration systems. Although these results seem predictable, they discovered that adenosine-5'-phosphosulfate (APS) can be directly utilized by sulfotransferase as a sulfonate donor, enabling a new and simple sulfonate group donor (APS) regeneration system from shortcutting the PAP to AMP reaction step. This would be a noteworthy result! However, the authors only demonstrated endpoint analysis of the reaction. The authors should examine how much efficiently sulfotransferase can utilize APS as a sulfonate donor compared to PAPS, i.e., K_m (and k_{cat}) of enzyme in the presence of different cofactors (PAPA and APS).

2, The authors isolated a mutant sulfotransferase that can utilize APS 1.5-fold efficiently than the wild type. Unfortunately, the increase does not seem progressive and sufficient in the industrial applications.

3, Furthermore, the authors mentioned that the concentration of polyP6 was optimized and finally determined to be 2.0 mM. The upper limit of sulfonate product concentrations must depend on those of substrates and regeneration system. Thus, this level would be insufficient concentration in the industrial applications.

4, P. Datta et al., reported unique PAPS regeneration system using p-nitrophenol sulfate (PNPS) as a

sulfate donor (Applied Microbiology and Biotechnology volume 104, pages 7067–7078 (2020)). The authors should discuss this technique.

Minor:

5, The authors termed “phosphorylase” toward the reverse reaction of “polyphosphate kinase”. In general terms, phosphorylases are enzymes that catalyze the addition of a phosphate group from an inorganic phosphate to an acceptor. Polyphosphate kinase reversibly transfers a phosphate group from a donor (polyphosphate) to an acceptor.

6, Line 127-128, 150, What is “active sulfonate group equivalent value”? What are “ATP equivalent values”? Please define or briefly explain them in the text.

7, Line 203-204, The authors mentioned “the constructed PAPS system is suitable for introduction into living cells”. How you introduce polyphosphate substrate into cell?

Reviewer #3:

Remarks to the Author:

Sulfation of various biological molecules is essential for bioactivation and detoxification. Since PAPS has been accepted as a unique active sulfonate group donor for all sulfation reaction in living organisms, an adequate supply of PAPS is indispensable.

In this manuscript, Xu et al. designed and constructed a de novo PAPS regeneration system that are assembled from the following three modules: the conversion of AMP from ATP, the biosynthesis of PAPS from ATP, and the regeneration of AMP from the accumulated PAP during PAPS consumption. In conjunction with utilization of the previously constructed, bifunctional PAPS synthase ASAK, optimal screening of the enzymes for modules 1 and 3 enabled the construction of the de novo system in this study. Furthermore, the authors found that APS, a precursor of PAPS, can also function as a universal active sulfonate group donor, and established a brand-new APS regeneration system.

Experiments are carefully performed and all data presented in this manuscript is convincing.

Therefore, this reviewer feels that the manuscript is acceptable in principle, but still requires some revisions.

Major points #1

How did the authors identify the structure of chondroitin disaccharide A? If the sulfation reaction by C4ST was conducted using chondroitin polymer, the sulfated chondroitin might be depolymerized with chondroitinase ABC. Since chondroitinase ABC is an eliminase but not a hydrolase, this treatment should generate chondroitin sulfate disaccharides that consist of unsaturated hexuronic acid and N-acetylgalactosamine with or without sulfonate groups. Therefore, the disaccharide structure in Supplementary Fig. 3B appears to be mistaken. For similar reasons, the structure of N-sulfated heparin disaccharide in Supplementary Fig. 5A also appears to be mistaken. The authors should clarify these points.

Major points #2

As shown in Figs. 5D and 5E, modification of the 3'PB motif of sulfotransferase may be promising for increased production of sulfonate group compounds in a condition using the APS-mediated sulfonate group regeneration system. However, relative production level of sulfated molecules in the APS-mediated regeneration system combined with the 3'PB motif-modified sulfotransferases toward that in the PAPS-mediated regeneration system is obscure. Therefore, the authors should clarify this point to emphasize the usability of this APS-mediated system.

Minor points #1

Supplementary Fig. 1B, data for the polymerization degree of polyphosphates for ADP phosphorylation were missing.

Minor points #2

Page 7, lines 136-138, Fig. 3D-G were miscited in the text. In addition, Fig. 3B was not cited in the text.

Minor points #3

Page 8, lines 155, "Fig. 5C" should be replaced with "Fig. 4B". In addition, explanation for Supplementary Fig. 9 (Page 8 lines 157-158) was incorrect.

Minor points #4

Supplementary References 3-5 were not cited in Supplementary information.

Reviewer #4:

Remarks to the Author:

In this study, the authors describe two sulfo donor regeneration systems, the PAPS regeneration system and the APS regeneration system, which aim to enhance the biosynthesis of sulfated compounds. The PAPS regeneration system consists of three modules that catalyze the cascade conversion of AMP to ATP, ATP to PAPS, and PAP to AMP, using sulfate and polyphosphates as substrates.

Major comments:

1, The authors need to demonstrate the superiority of the current PAPS regeneration system over existing ones by setting up an appropriate control group. For instance, they could compare the sulfonation degree and sulfonate group equivalence values achieved using the current PAPS regeneration system with those obtained using the AST IV-based co-factor recycling system (Burkart et al. 2000; Bhaskar et al. 2015). Comparing the current system with an incomplete system that lacks some components, as shown in Fig. 2D and 2E, is insufficient to prove its efficiency.

2, The authors need to demonstrate the wide applicability of the PAPS regeneration system through experiments. Many sulfotransferases require Ca^{2+} , Mn^{2+} , Mg^{2+} , etc. (Xu et al. 2011), but polyP6 interacts with these divalent cations, forming large amounts of precipitate that prevents the catalytic reaction from proceeding properly. Therefore, the PAPS regeneration system including polyP6 cannot be coupled with these valuable sulfation reactions, which greatly limits its applicability.

3, Cost is a key consideration when it comes to the sulfation of organic compounds, and the authors need to consider various factors to objectively reflect the real cost. The authors introduce the enzyme CysQ to decompose PAP to form AMP, then express the enzyme PPK2 to convert AMP into ADP and ATP, and finally use three enzymes of module 2 to convert ATP into PAPS. In contrast, in the AST IV-based co-factor recycling system, only one enzyme is used to directly convert PAP into PAPS, using a 5 mM concentration of PNPS as the sulfogroup donor (Burkart et al. 2000; Bhaskar et al. 2015). Therefore, in the cost calculation of Supplementary Table 4, a comprehensive consideration of various factors, such as the cultivation of bacteria, enzyme expression and purification, enzyme production yield and catalytic efficiency, and time and personnel costs, is necessary to determine the real reaction cost.

4, polyP6's high density of negative charges will interfere with the purification of PAPS by a DEAE column, making it very difficult to purify the PAPS synthesized by this system.

Minor comments:

1, As shown in Fig. 3, the sulfation efficiency of APS as a sulfo donor is much lower than that of PAPS,

and hence, the authors should move the studies about APS to the supplementary section.

2, Although sulfotransferases can be inhibited by PAP, this is not an insurmountable problem. In previous sulfation studies, up to 600 μM of PAPS was added to the enzymatic reaction solution (Xu et al. 2011). Therefore, a more complete statement should be made regarding the issue of PAP inhibition in the Discussion section.

3, The authors also need to clarify some vague sentences, such as those on page 2 lines 24-27 and page 3 lines 36-38.

4, The word "values" on page 7 line 123 is repeated and should be removed.

Bhaskar U et al. (2015) Combinatorial one-pot chemoenzymatic synthesis of heparin. *Carbohydrate Polymers* 122:399-407 doi: <http://dx.doi.org/10.1016/j.carbpol.2014.10.054>

Burkart MD, Izumi M, Chapman E, Lin CH, Wong CH (2000) Regeneration of PAPS for the enzymatic synthesis of sulfated oligosaccharides. *J Org Chem* 65:5565-5574

Xu Y et al. (2011) Chemoenzymatic synthesis of homogeneous ultralow molecular weight heparins. *Science* 334:498-501 doi: [10.1126/science.1207478](https://doi.org/10.1126/science.1207478)

Detailed response to the reviewers' comments:

Reviewer #1 (Remarks to the Author):

“Xu et al. introduce a novel 3'-phosphoadenosine-5'-phosphosulfate (PAPS) regeneration system that encompasses modules responsible for converting AMP to ATP, ATP to PAPS, and PAPS to AMP using sulfotransferase. This PAPS and APS biosynthesis system was employed to produce a range of sulfated compounds. Although I found the study intriguing, it seems too preliminary for publication at this stage.”

Author's response: We appreciate the reviewer for the evaluations and helpful comments. According to the comments, we have conducted more experiments, and added more significant results in the revised manuscript to increase its significance.

“Firstly, while the *in vitro* production of sulfated products using PAPS and APS regeneration systems appeared impressive, the authors need to demonstrate the superiority of their system within cellular biosynthesis (e.g., by expressing all necessary enzymes and examining the production yield of biosynthesized sulfated products).”

Author's response: We thank the reviewer for this good suggestion. Accordingly, we have also introduced this PAPS regeneration system into *E. coli* cell and demonstrated its superiority for cellular biosynthesis. Briefly, we firstly demonstrated that *E. coli* is able to uptake the polyP₆ to increase intracellular ATP concentration. Then, we engineered an *E. coli* strain by overexpressing PAPS bifunctional synthase ASAK, polyphosphokinase PPK2^{s,e}, PAP 3'-dephosphatase *KpCysQ* and SULTs. Based on this strain, we applied phenol (aryl) sulfotransferase AST IV as an example to produce the natural anti-adhesive agent, zosteric acid. The production of zosteric acid increased by 14.3 folds to 1.51 g/L after introducing PAPS regeneration system (Figure 3). In consideration of the significance, we have added these results in the revised manuscript. We appreciate the reviewer for this good suggestion.

“Additionally, several PAPS regeneration systems have been reported in previous studies (e.g., *Biotechnol. J.* 2019, 14, e1800436; *J. Bacteriol.* 1992, 174, 415). The authors should compare the efficiencies of these systems with their engineered system and clearly outline the advantages of their approach over existing systems.”

Author’s response: As the reviewer directed, Badri A *et al.* engineered a PAPS-accumulating strain to synthesize chondroitin sulfate A. The engineered *E. coli* strain showed a 1000-fold remarkable increase in intracellular PAPS concentrations. However, due to the lack of direct phosphate donors, the use of PAPS is disposable. Therefore, their PAPS accumulation strategy afforded only the transformation of 5 µg chondroitin to chondroitin sulfate, with 37% conversion rate. (*Biotechnol. J.* 2019, 14, e1800436). Neuwald AF *et al.* reported that 3'-phosphoadenoside 5'-phosphosulfate is toxic if allowed to accumulate, and found that CysQ help control the pool of 3'-phosphoadenoside 5'-phosphosulfate. This result provides an important reference for our construction of intracellular PAPS regeneration system. We have added the necessary discussions in the revised manuscript. (*J. Bacteriol.* 1992, 174, 415). As suggested, in order to comprehensively compare the present PAPS regeneration system with previously reported ones, we have added Supplementary Table S3 in the supplemental information. We appreciate the reviewer for this good suggestion.

References:

1. Badri A. *et al.* Increased 3'-phosphoadenosine-5'-phosphosulfate levels in Engineered *Escherichia coli* cell lysate facilitate the *in vitro* synthesis of chondroitin sulfate A. *Biotechnol J.* 2019;14(9):e1800436.
2. Neuwald AF. *et al.* *cysQ*, a gene needed for cysteine synthesis in *Escherichia coli* K-12 only during aerobic growth. *J Bacteriol.* 1992;174(2):415-25.

“Finally, the authors semi-rationally engineered Stf0 to enhance its APS affinity. Why was Stf0 chosen as a target? Did the authors attempt to work with any of the other sulfotransferases mentioned in this study?”

Author’s response: It is a good question and suggestion. We firstly chosen Stf0 as a

target due to the 3D structure of Stf0 has been solved (PDB ID: 1TEX). In this regard, we could precisely determine the position of 3'-PB motif. As the reviewer suggested, in order to expand the application scenarios, we have further modified the rat phenol sulfotransferase AST IV. Because of the lack of 3D structure, the definite 3'-PB motif region was determined by sequence alignment. Our results also proved that the 3'-PB motif region has a very important degree of freedom for improving APS affinity (Figure 7).

I also have a few minor comments: “1. Please include chemical structures in Figure 1.”

Author’s response: We have added chemical structures in Figure 1.

“2. In Figure 1A, compare the engineered system with natural systems.”

Author’s response: As suggested, we have added the natural PAPS biosynthesis pathway and the possible enzymes that involve in its regeneration in Figure 1A.

“3. Could the authors indicate the reaction time length in Figure 1B?”

Author’s response: It’s a good suggestion, we have indicated the reaction time length in Figure 1B.

“4. On line 109, correct the formatting error with "(Fig. 1F" by removing the bold formatting from the opening parenthesis.”

Author’s response: We have corrected the formatting error in the revised manuscript.

Reviewer #2 (Remarks to the Author):

“To construct *in vitro* biochemical sulfonation system, the authors developed phosphoadenosine-5'-phosphosulfate (PAPS, sulfonate group donor) regeneration systems composed of three modules; module 1, ATP regeneration from AMP using polyphosphate kinase; module 2, PAPS synthesis using PAPS synthetase; module 3, transferring sulfonate group from PAPS to target compounds using sulfotransferase and dephosphorylation of phosphoadenosine-5'-phosphate (PAP) to AMP. They demonstrated chondroitin sulfate and trehalose sulfate synthesis using the PAPS regeneration systems. Although these results seem predictable, they discovered that adenosine-5'-phosphosulfate (APS) can be directly utilized by sulfotransferase as a sulfonate donor, enabling a new and simple sulfonate group donor (APS) regeneration system from shortcutting the PAP to AMP reaction step. This would be a noteworthy result! However, the authors only demonstrated endpoint analysis of the reaction. The authors should examine how much efficiently sulfotransferase can utilize APS as a sulfonate donor compared to PAPS, i.e., K_m (and k_{cat}) of enzyme in the presence of different cofactors (PAPS and APS).”

Author's response: We appreciate the reviewer for the positive evaluations and suggestions on our manuscript. We have added experimental results regarding to the values of K_m , k_{cat} and k_{cat}/K_m (Table 1) to examine how much efficiently sulfotransferases can utilize APS as a sulfonate donor. We also believe that this part of the content provides a good reference for the follow-up research.

Enzymes	Substrates	K_m (mM)	k_{cat} (h ⁻¹)	k_{cat}/K_m (h ⁻¹ ·mM ⁻¹)
Stf0	APS	1.56 ± 0.26	21.24 ± 1.44	13.62
	PAPS	0.82 ± 0.06	35.58 ± 0.90	43.39
NST	APS	2.72 ± 0.41	47.02 ± 1.43	17.29
	PAPS	1.14 ± 0.15	72.50 ± 0.91	63.60
C4ST	APS	2.65 ± 0.52	21.86 ± 2.15	8.25
	PAPS	1.27 ± 0.14	55.02 ± 2.30	43.32
AST IV	APS	2.45 ± 0.50	24.78 ± 2.46	10.11
	PAPS	1.43 ± 0.172	47.3 ± 2.21	33.08
EST	APS	2.91 ± 0.52	25.56 ± 2.41	8.78
	PAPS	1.56 ± 0.26	30.58 ± 2.06	19.60

“2, The authors isolated a mutant sulfotransferase that can utilize APS 1.5-fold efficiently than the wild type. Unfortunately, the increase does not seem progressive and sufficient in the industrial applications.”

Author’s response: As the reviewer directed, the 1.5-fold increase in affinity make not indicate more promising industrial applications, but it gives us the right direction to continue engineering the 3'PB motif to enhance its affinity towards APS. Meanwhile, our identification and modification of the 3'PB motif paved the way for engineering SULTs. In order to further facilitate subsequent research, we have designed and conducted more experiments and successfully constructed a SULTs activity indicator system for special using APS, and applied it into the reprogramming of Stf0 and AST IV (see Figure 6 in the revised manuscript). The SULTs activity indicator established here could be applicable for other sulfotransferases, which will assist researchers to improve the APS affinity of other sulfotransferases of interest.

“3, Furthermore, the authors mentioned that the concentration of polyP₆ was optimized and finally determined to be 2.0 mM. The upper limit of sulfonate product concentrations must depend on those of substrates and regeneration system. Thus, this level would be insufficient concentration in the industrial applications.”

Author’s response: It is a valuable question. In our study, we found that addition of excess polyP₆ inhibits the catalytic process, 2.0 mM polyP₆ satisfied the current catalytic reaction. Thus, for future industrial applications, polyP₆ could be constantly feeding with a concentration of about 2.0 mM. Also, an alternative way is to decrease the inhibition and improve the reaction flux by engineering the enzyme of ATP sulfurylase that catalyze the conversion of ATP to APS.

“4, P. Datta *et al.*, reported unique PAPS regeneration system using *p*-nitrophenol sulfate (PNPS) as a sulfate donor (Applied Microbiology and Biotechnology volume 104, pages 7067–7078 (2020)). The authors should discuss this technique.”

Author's response: P. Datta *et al.* reported a unique PAPS synthesis and regeneration system. First, the 5'-phosphosulfate kinase (APSK), ATP sulfurylase (ATPS) and pyrophosphatase (PPA) were used for PAPS synthesis. Next, The PAPS was used for the chemoenzymatic synthesis of the heparan sulfate polysaccharide, during which reaction process the PAPS were regenerated via a PAPS-AST IV- *p*-nitrophenol sulfate (PNPS) regeneration system. The use of PNPS as the sulfate donor (this study, SO_4^{2-}), not only will lead to an increase in the price of the entire system, but it will also lead to the accumulation of PNP (toxic). As suggested, we have included the discussion on this technique in the revised manuscript (line 268-276). Meanwhile, according to other reviewers' comments, we compared different PAPS regeneration systems from various dimensions and provided a new Supplementary Table S3.

Reference:

1: Datta, P. *et al.* Expression of enzymes for 3'-phosphoadenosine-5'-phosphosulfate (PAPS) biosynthesis and their preparation for PAPS synthesis and regeneration. *Appl Microbiol Biotechnol.* 2020;104(16):7067-7078.

Minor:

“5. The authors termed “phosphorylase” toward the reverse reaction of “polyphosphate kinase”. In general terms, phosphorylases are enzymes that catalyze the addition of a phosphate group from an inorganic phosphate to an acceptor. Polyphosphate kinase reversibly transfers a phosphate group from a donor (polyphosphate) to an acceptor. ”

Author's response: We apologize for the ambiguity caused by our descriptions. Our opinion is consistent with that of the reviewer. AD(M)P phosphorylase means catalyze the addition of a phosphate group to AD(M)P. We carefully revised the description of “phosphorylase” in the manuscript. We are very grateful for the reviewer's correction.

“6, Line 127-128, 150, What is “active sulfonate group equivalent value”? What are

“ATP equivalent values”? Please define or briefly explain them in the text.”

Author’s response: Thank reviewer for ask good questions. The amount of sulfonate groups transferring to substrate that driven by initial equal ATP was defined as “the active sulfonate group equivalent value or ATP equivalent value”. This definition excludes which sulfonate group donor (PAPS/APS) is used and can characterize the efficiency of the constructed PAPS or APS regeneration system. To avoid misunderstanding, we removed the statement of “ATP equivalent values” and revised the explanation of “the active sulfonate group equivalent (ASGE) values” in the revised manuscript. We appreciate the reviewer for these good comments and suggestions.

“7, Line 203-204, The authors mentioned “the constructed PAPS system is suitable for introduction into living cells”. How you introduce polyphosphate substrate into cell?”

Author’s response: It is a good question. Recently, the study for construction of ATP regeneration system in *Streptomyces albulus* (*Microb Cell Fact.* 2023. 22(1):51.) has reported that polyP₆ was added and absorbed by *S. albulus*. Here, also as the other reviewer suggested, we tried to introduce this PAPS regeneration system into living cell. So firstly we investigated whether *E. coli* could use the extracellular polyP₆. Our results confirmed that when adding polyP₆ to the culture medium, the content of ATP in *E. coli* cells was significantly increased (Figure 3), demonstrating *E. coli* is able to uptake polyP₆. In view of the significance of this result, we have added this part in the revised manuscript.

Reference:

1. Yang H. *et al.* Engineering *Streptomyces albulus* to enhance ϵ -poly-L-lysine production by introducing a polyphosphate kinase-mediated ATP regeneration system. *Microb Cell Fact.* 2023; 14;22(1):51.

Reviewer #3 (Remarks to the Author):

“Sulfation of various biological molecules is essential for bioactivation and detoxification. Since PAPS has been accepted as a unique active sulfonate group donor for all sulfation reaction in living organisms, an adequate supply of PAPS is indispensable. In this manuscript, Xu et al. designed and constructed a de novo PAPS regeneration system that are assembled from the following three modules: the conversion of AMP from ATP, the biosynthesis of PAPS from ATP, and the regeneration of AMP from the accumulated PAP during PAPS consumption. In conjunction with utilization of the previously constructed, bifunctional PAPS synthase ASAK, optimal screening of the enzymes for modules 1 and 3 enabled the construction of the de novo system in this study. Furthermore, the authors found that APS, a precursor of PAPS, can also function as a universal active sulfonate group donor, and established a brand-new APS regeneration system. Experiments are carefully performed and all data presented in this manuscript is convincing. Therefore, this reviewer feels that the manuscript is acceptable in principle, but still requires some revisions.”

Author’s response: We appreciate the reviewer for the positive evaluations and suggestions on our manuscript. According to the comments, we have carefully revised the manuscript. Also, more experiments have been carried out and new significant results have been added in the revised manuscript.

Major points #1

“How did the authors identify the structure of chondroitin disaccharide A? If the sulfation reaction by C4ST was conducted using chondroitin polymer, the sulfated chondroitin might be depolymerized with chondroitinase ABC. Since chondroitinase ABC is an eliminase but not a hydrolase, this treatment should generate chondroitin sulfate disaccharides that consist of unsaturated hexuronic acid and N-acetylgalactosamine with or without sulfonate groups. Therefore, the disaccharide structure in Supplementary Fig. 3B appears to be mistaken. For similar reasons, the

structure of N-sulfated heparin disaccharide in Supplementary Fig. 5A also appears to be mistaken. The authors should clarify these points.”

Author’s response: Here, a chondroitinase ABC I was used for identification and quantification of chondroitin sulfate disaccharides. Actually, as the reviewer directed, the disaccharides contain unsaturated hexuronic acid. We have modified the chemical formula structure in the Figure 4D and 4E. We thank the reviewer for this reminding.

Major points #2

“As shown in Figs. 5D and 5E, modification of the 3’PB motif of sulfotransferase may be promising for increased production of sulfonate group compounds in a condition using the APS-mediated sulfonate group regeneration system. However, relative production level of sulfated molecules in the APS-mediated regeneration system combined with the 3’PB motif-modified sulfotransferases toward that in the PAPS-mediated regeneration system is obscure. Therefore, the authors should clarify this point to emphasize the usability of this APS-mediated system.”

Author’s response: Here, to compare the efficiency of different systems for providing sulfonate group, we defined “the active sulfonate group equivalent (ASGE) value” as the sulfonation efficiency, which means the amount of sulfonate groups transferring to substrate that driven by initial equal ATP. Accordingly, the ASGE values of the constructed PAPS or APS regeneration systems were characterized and compared. For enzymatic production of trehalose-2-sulfate, the ASGE value is 3.82 when using the PAPS regeneration system (Figure 2E). In contrast, the active sulfonate group equivalent value was only 2.18 with wild type Stf0 for the APS regeneration system. After engineering the 3’PB motif of sulfotransferase Stf0, the active sulfonate group equivalent value was increased to 3.26 (Fig. 7G), demonstrating the usability of this APS-mediated system. In the revised manuscript, we added a rapid system for analyzing the activity of sulfotransferase, which would facilitate the rapid optimization of the APS system with specific sulfotransferase. We appreciate the reviewer for the

good suggestion.

Minor points #1

“Supplementary Fig. 1B, data for the polymerization degree of polyphosphates for ADP phosphorylation were missing.”

Author’s response: Thank reviewer for this reminding. We have added the polymerization degree of polyphosphates in Supplementary Fig. 1B.

Minor points #2

“Page 7, lines 136-138, Fig. 3D-G were miscited in the text. In addition, Fig. 3B was not cited in the text.”

Author’s response: Thank reviewer for the carefulness. We have recited Fig. 3D-G and Fig. 3B, and carefully checked all the figures in the revised manuscript.

Minor points #3

“Page 8, lines 155, “Fig. 5C” should be replaced with “Fig. 4B”. In addition, explanation for Supplementary Fig. 9 (Page 8 lines 157-158) was incorrect.”

Author’s response: We have replaced “Fig. 5C” with “Fig. 4B”, and corrected the explanation for Supplementary Fig. 9.

Minor points #4

“Supplementary References 3-5 were not cited in Supplementary information.”

Author’s response: We have cited the supplementary references 3 (Supplementary Table S1), 4 (Supplementary Table S1) and 5 (Supplementary Table S3) in Supplementary information, and carefully checked the attachment material information. We sincerely thank the reviewer for careful reading.

Reviewer #4 (Remarks to the Author):

“In this study, the authors describe two sulfo donor regeneration systems, the PAPS regeneration system and the APS regeneration system, which aim to enhance the biosynthesis of sulfated compounds. The PAPS regeneration system consists of three modules that catalyze the cascade conversion of AMP to ATP, ATP to PAPS, and PAP to AMP, using sulfate and polyphosphates as substrates.”

Author’s response: We would like to thank the reviewer for the constructive suggestions, which has significantly improved the presentation of our manuscript.

Major comments:

“1, The authors need to demonstrate the superiority of the current PAPS regeneration system over existing ones by setting up an appropriate control group. For instance, they could compare the sulfonation degree and sulfonate group equivalence values achieved using the current PAPS regeneration system with those obtained using the AST IV-based co-factor recycling system (Burkart et al. 2000; Bhaskar et al. 2015). Comparing the current system with an incomplete system that lacks some components, as shown in Fig. 2D and 2E, is insufficient to prove its efficiency.”

Author’s response: We thank the reviewer for this good question. As suggested, we have compared the reported different PAPS regeneration systems in detail in the new Supplementary Table S3, and compared the sulfonation degree achieved by these PAPS regeneration systems. Fig. 2 depicted the development process of the PAPS regeneration system of our study. To this end, module I (the biosynthesis of ATP from AMP) and module III (the regeneration of AMP from PAPS) were investigated and characterized in Fig. 2BCD and Fig. 2E, respectively. In our previous study (*ACS Catal.* 2021.16:10405–10415), module II has been constructed and characterized. In module 3, the committed reaction is the conversion of PAP to AMP. After characterizing and optimizing all the other reactions in the PAPS regeneration system, the rest thing for researchers is to select specific sulfotransferase of interest (in module III) for specific

sulfonation modification.

Here, our results showed that the built modules can work independently and can be assembled together for sulfonation modification with specific sulfotransferase. Obviously, the sulfonation efficiency was significantly increased by applying this PAPS regeneration system (Fig. 2C and 2E in the revised manuscript). To avoid misunderstanding, the description of “After engineering the PAPS regeneration system, the contribution of modules 1 and 3 to sulfonation was investigated” was modified to “After characterizing and optimizing the committed enzymes for module I and module III, the sulfonation efficiency of the PAPS regeneration system was compared with that of the incomplete sulfonation systems” in the revised manuscript.

Reference:

Xu, R et al. Closed-loop system driven by ADP phosphorylation from pyrophosphate affords equimolar transformation of ATP to 3'-phosphoadenosine-5'-phosphosulfate. 2021. *ACS Catal.* 2021. 16, 10405–10415.

“2, The authors need to demonstrate the wide applicability of the PAPS regeneration system through experiments. Many sulfotransferases require Ca^{2+} , Mn^{2+} , Mg^{2+} , etc. (Xu et al. 2011), but polyP₆ interacts with these divalent cations, forming large amounts of precipitate that prevents the catalytic reaction from proceeding properly. Therefore, the PAPS regeneration system including polyP₆ cannot be coupled with these valuable sulfation reactions, which greatly limits its applicability.”

Author's response: It is a good question. We agree the reviewer that high concentration of polyP₆ resulted in the sedimentation of divalent cations. As the reviewer directed, we also found that the addition of Mg^{2+} was positive for the high activity of sulfotransferase C4ST in our previous studies (*Green Chem.* 2022; 24, 3180-3192, *Biotechnol. Bioeng* 2018; 115, 1561-1570). Thus, to avoid precipitation, we have optimized the concentration of polyP₆ to be 2.0 mM. In this case, no precipitation was

discovered. For industrial applications, a constant feeding strategy could be used for the guarantee of appropriate concentration of polyP₆ in the reaction system.

References:

1. Zhou, Z. et al. A microbial-enzymatic strategy for producing chondroitin sulfate glycosaminoglycans. *Biotechnol. Bioeng* 115, 1561-1570 (2018).
2. Zhang, Y. et al. Synthesis of bioengineered heparin by recombinant yeast *Pichia pastoris*. *Green Chem.* 24, 3180-3192 (2022).

“3, Cost is a key consideration when it comes to the sulfation of organic compounds, and the authors need to consider various factors to objectively reflect the real cost. The authors introduce the enzyme CysQ to decompose PAP to form AMP, then express the enzyme PPK2 to convert AMP into ADP and ATP, and finally use three enzymes of module 2 to convert ATP into PAPS. In contrast, in the AST IV-based co-factor recycling system, only one enzyme is used to directly convert PAP into PAPS, using a 5 mM concentration of PNPS as the sulfogroup donor (Burkart et al. 2000; Bhaskar et al. 2015). Therefore, in the cost calculation of Supplementary Table S3, a comprehensive consideration of various factors, such as the cultivation of bacteria, enzyme expression and purification, enzyme production yield and catalytic efficiency, and time and personnel costs, is necessary to determine the real reaction cost.”

Author’s response: We agree with the reviewer's comments that cost calculation should consider various factors. In consideration of the complexity for calculating the exact cost of each system, we modified Supplementary Table S3 to compare the advantages of these PAPS regeneration system. In comparison, the obvious advantages for the present PAPS regeneration system are: (1) no addition of substrates such as PEP and PNPS with high cost; (2) no toxic intermediate products such as PNP was accumulated in the sulfonation system.

“4, PolyP₆'s high density of negative charges will interfere with the purification of PAPS by a DEAE column, making it very difficult to purify the PAPS synthesized by this system.”

Author's response: We appreciate the reviewer for this insightful comment. As the reviewer directed, in our previous study we indeed found that polyP₆ affect the purification of PAPS. Thus, pyrophosphoric acid (PPi) was selected as the phosphate group donor for PAPS biosynthesis. In this study, our aim is to design and construct a PAPS regeneration system. In this case, no PAPS purification step was needed in the system. As a result, to facilitate the operation of the PAPS regeneration system, PolyP₆ was used as phosphate group donor.

Minor comments:

“1, As shown in Fig. 3, the sulfation efficiency of APS as a sulfo donor is much lower than that of PAPS, and hence, the authors should move the studies about APS to the supplementary section.”

Author's response: In this study, we discovered that in addition to PAPS, APS is also a new sulfonate donor. According to the comments from other reviewers, we have carried out new experiments and added the results in the revised manuscript. Although the sulfation efficiency for APS with the wild type sulfotransferase is lower than that of PAPS, we achieved the compatible value after engineering the region for APS binding. As an example, for enzymatic production of trehalose-2-sulfate, the active sulfonate group equivalent value is 3.82 when using the PAPS regeneration system (Figure 2E). After engineering the 3'PB motif of sulfotransferase Stf0, the active sulfonate group equivalent value was increased from 2.18 to 3.26 (Fig. 7H, 7I), demonstrating the usability of this APS-mediated system. Also, a rapid system for analyzing the activity of sulfotransferase was added in the revised manuscript, which would facilitate the rapid optimization of the APS system with specific sulfotransferase. We appreciate the reviewer for the good suggestion.

“2, Although sulfotransferases can be inhibited by PAP, this is not an insurmountable problem. In previous sulfation studies, up to 600 μ M of PAPS was added to the enzymatic reaction solution (Xu et al. 2011). Therefore, a more complete statement should be made regarding the issue of PAP inhibition in the Discussion section.”

Author’s response: Many thanks to the reviewer for this good suggestion. “Since the structures of PAPS and PAP are very similar, they were found to have similar affinities for the enzymes (*Angew Chem Int Ed Engl.* 2004. 43:3526-3548). The continued occupation of PAP in the sulfotransferase pocket resulted in the prevention of PAPS binding. Thus, the presence of PAP, especially with a high concentration, should affect PAPS utilization. As the reviewer directed, Xu et al. applied 600 μ M of PAPS for chemoenzymatic synthesis of homogeneous ultralow molecular weight heparins, suggesting the accumulation of PAP with a low concentration generated insignificant inhibition on sulfonation (*Science.* 2011. 334:498-501). Even so, construction of a PAPS regeneration system without accumulation of PAP should be more preferable. In the revised manuscript, we have added more discussion for PAP inhibition.

References:

1. Chapman E. et al. Sulfotransferases: structure, mechanism, biological activity, inhibition, and synthetic utility. *Angew Chem Int Ed Engl.* 2004. 43(27):3526-3548.
2. Xu, Y. et al. Chemoenzymatic synthesis of homogeneous ultralow molecular weight heparins. *Science.* 2011. 334: 498-501.

“3, The authors also need to clarify some vague sentences, such as those on page 2 lines 24-27 and page 3 lines 36-38.”

Author’s response: Thank reviewer for the suggestions, we have modified the sentences to make it easy to understand in the revised manuscript.

“4, The word "values" on page 7 line 123 is repeated and should be removed.”

Author’s response We have removed the word "values" on page 7 line 123.

Reviewers' Comments:

Reviewer #1:

Remarks to the Author:

The authors have provided a comprehensive response to the reviewers' comments and have significantly improved the manuscript. Reviewer 1 and 2's comments have been thoroughly addressed. However, before publishing in nature communications, the manuscript requires significant polishing.

Additionally, the following comments should be considered:

- 1) The authors should justify the claim "because of the low efficiencies of paps biosynthesis and paps regeneration." based on the new supporting figure 3, it seems that toxicity is the limitation, not the efficiency.
- 2) The manuscript requires substantial improvement in its language, for example, "first, we assembled... And also demonstrate..." on page 2, and "sulfation is essential for.. Are changed."
- 3) In the legend of figures, authors should include the full names of all abbreviations.
- 4) The sequences of the final plasmids for aps biosynthesis should be included in the supporting information.
- 5) "pas" on page 3 should be spelled out.
- 6) A reference should be added to "paps biosynthesis and decomposition..." on page 5.
- 7) On page 7, the csa (chondroitin sulfate a) abbreviation should be placed in parentheses.
- 8) What does the question mark stand for in fig. 3a?
- 9) On page 12, arginine and serine residues should be written out as "arginine and serine" instead of using "r and s." one-letter abbreviations should only be used for mutations.
- 10) The chemical structures in figure 1 are too small.

Please make these revisions and improvements to enhance the overall quality of the manuscript before resubmitting it for publication.

Reviewer #3:

Remarks to the Author:

The revised manuscript has been considerably improved by the additions and corrections made in response to reviews' comments. I have no more requirements.

Reviewer #4:

Remarks to the Author:

1: No comparative results were found regarding the sulfonation degree and higher sulfonate group equivalence values achieved using the current PAPS regeneration system in comparison to those obtained using the AST IV-based co-factor recycling system (Burkart et al., 2000; Bhaskar et al., 2015).

2:

Several critical sulfotransferases involved in ULMW heparin synthesis rely on the presence of divalent cations such as Ca^{2+} , Mn^{2+} , and Mg^{2+} at concentrations ranging from 2 to 10 mM. It is crucial to exclude any phosphate from the reaction mixture when using divalent cations, especially Ca^{2+} and Mn^{2+} . Including 2 mM polyP6 in the reaction mixture results in a comparable amount of phosphate groups from about 10 mM phosphate solution, which is high enough to reduce the concentration of free divalent cations. Consequently, the PAPS regeneration system that incorporates polyP6 cannot be effectively combined with these sulfation reactions, significantly limiting its practicality. Unfortunately, the authors did not provide a satisfactory explanation or solution to address this limitation.

3:

The authors employed five enzymes to perform the catalytic tasks a single enzyme could achieve. However, they did not provide a compelling side-by-side comparison demonstrating that when combined as a catalytic unit, the five enzymes offer superior accessibility and higher catalytic efficiency than the single enzyme (AST IV). For instance, they did not compare the yield and cost of sulfation by substituting AST IV in the No. 1 reaction with the five enzymes utilized in the No. 3 reaction. This lack of comparison raises questions about the advantages and effectiveness of employing multiple enzymes in this context.

In sulfation reactions utilizing AST IV for PAPS recycling, the addition of a catalytic amount of PAPS is adequate instead of using expensive PAP (refer to Table S3, reaction No. 1). Once the sulfo group of PAPS is transferred to the substrate, PAP is naturally generated. Moreover, recent studies have addressed the cost of PAPS. Therefore, the cost calculation related to PAP may not be valid (refer to Table S3).

4:

The high density of negative charges in PolyP6 presents a considerable obstacle not only for the purification of PAPS but also for the purification of any sulfated products obtained through the current PAPS regeneration system. PolyP6 is expected to cause significant interference during the purification process, particularly when utilizing anion exchange resins, which are commonly employed in sulfo modification.

Reviewers' comments

Reviewer #1 (Remarks to the Author):

“The authors have provided a comprehensive response to the reviewers' comments and have significantly improved the manuscript. Reviewer 1 and 2's comments have been thoroughly addressed. However, before publishing in nature communications, the manuscript requires significant polishing.”

Author's response: We appreciate the reviewer for the positive evaluation and comments on our manuscript. As suggested, we have corrected the corresponding issues and carefully revised the manuscript.

“Additionally, the following comments should be considered:

1) The authors should justify the claim "because of the low efficiencies of PAPS biosynthesis and PAPS regeneration." based on the new supporting Figure 3, it seems that toxicity is the limitation, not the efficiency.”

Author's response: Thanks for the reviewer's suggestion, we have justified the corresponding statements in the revised the manuscript.

“2) The manuscript requires substantial improvement in its language, for example, "first, we assembled... And also demonstrate..." on page 2, and "sulfation is essential for... Are changed.”

Author's response: We have modified the above sentences and have carefully revised the manuscript and do our best to polish the language of our manuscript.

“3) In the legend of figures, authors should include the full names of all abbreviations.”

Author's response: Thanks for the reviewer's suggestion. We have included the full names of all abbreviations in the legend of figures.

“4) The sequences of the final plasmids for APS biosynthesis should be included in the supporting information.”

Author's response: We have included the sequences of the final plasmids for APS biosynthesis in the supporting information.

“5) "PAS" on page 3 should be spelled out.”

Author's response: We have corrected this error.

“6) A reference should be added to "PAPS biosynthesis and decomposition..." on page 5.”

Author's response: Thanks for the reviewer's suggestion, we have added a new reference “34. Günal, S., Hardman, R., Kopriva, S. & Mueller, J.W. Sulfation pathways from red to green. *J. Biol. Chem.* 294, 12293-12312 (2019).” to “PAPS biosynthesis and decomposition...”.

“7) On page 7, the CSA (chondroitin sulfate A) abbreviation should be placed in parentheses.”

Author’s response: We have corrected this error.

“8) What does the question mark stand for in Fig. 3A?”

Author’s response: The question mark means that it is not known whether polyphosphate can enter the cell at the beginning. To reduce misunderstandings, we have removed the question mark from the Fig. 3A.

“9) On page 12, arginine and serine residues should be written out as "arginine and serine" instead of using "r and s." one-letter abbreviations should only be used for mutations.”

Author’s response: we have written out arginine and serine residues as "arginine and serine".

“10) The chemical structures in Figure 1 are too small. Please make these revisions and improvements to enhance the overall quality of the manuscript before resubmitting it for publication.”

Author’s response: Thanks for the reviewer's suggestion. We have enlarged the chemical structures.

Reviewer #3 (Remarks to the Author):

“The revised manuscript has been considerably improved by the additions and corrections made in response to reviews' comments. I have no more requirements.”

Author’s response: Thanks to the feedback and comments provided by the reviewer, which allowed we to make so many improvements and helped to improve the quality of the manuscript.

Reviewer #4 (Remarks to the Author):

“1: No comparative results were found regarding the sulfonation degree and higher sulfonate group equivalence values achieved using the current PAPS regeneration system in comparison to those obtained using the AST IV-based co-factor recycling system (Burkart et al., 2000; Bhaskar et al., 2015).”

Author's response:

In fact, we have used the AST IV-based co-factor recycling system in our previous studies (Zhou Z *et al. Biotechnol Bioeng.* 2018. 115:1561-1570. Xi X, *et.al. J. Ind. Microbiol. Biotechnol.* 2023 17;50: kuad012.). Initially, we thought that comparisons with the data from literatures are sufficient and did not use our own experiment data. As you suggested, we have added the results of AST IV-based co-factor recycling system in the revised manuscript and compared it with the constructed PAPS regeneration system (Fig. 2F, Supplementary Fig. 5). Obviously, when sulfonating substrate with same concentration (5 g/L trehalose), the PAPS regeneration system with polyP₆ (energy source) and SO₄²⁻ (sulfonate donor) achieved higher sulfonation rate and efficiency comparing with AST IV-based co-factor recycling system (Please see Fig. 1 below). We appreciate reviewer 4 for this suggestion. Moreover, as suggested from other three reviewers, in view of the low efficiency of the APS regeneration system, we designed and constructed a rapid APS (adenosine 5'-phosphosulfate)-mediated sulfotransferase activity indicating system for screening sulfotransferase variants with high utilization efficiency towards APS. Eventually, we engineered efficient APS regeneration systems by coupling sulfotransferase variants (three enzymes in total), which achieved similar sulfonation efficiency (3.26 ASGE values, 5 g/L trehalose, 85% yield) comparing to the PAPS regeneration system. In comparison, the APS regeneration system with shorter route is more applicable to be scaled up.

In consideration of different enzymes used in different PAPS regeneration system, it is rational to compare the price of substrates from the perspective of practical applications. The high price of the sulfonate donor PNPS (10-times higher than ATP) and the accumulation of the toxic compound PNP encouraged us to develop novel green sulfonate donor regeneration systems by integrating sulfotransferase with inexpensive substrates polyP₆ and SO₄²⁻. The applicability of using polyP₆ as phosphate donor has been demonstrated in our previous study for PAPS biosynthesis (Xu *et al. ACS Catal.* 2021. 11:10405-415). In fact, polyP₆ and polyP_n (even longer than 6) has been widely used in polyphosphokinase-mediated ATP regeneration system to synthesize various food products and drugs recently (Cai *et al. Science.* 2021. 373:1523-27; Tavanti *et al. Green Chem.* 2021. 23: 828-37; Petchey *et al. ACS Catal.* 2020. 10: 4659-63). In future, we plan to use the protein crystalline inclusion-based enzyme immobilization systems (Wang *et al. ACS Synth Biol.* 2023.12:1487-96) to immobilize the enzymes of the PAPS and APS regeneration system.

Fig. 1. Final sulfonation intensity at different conditions (A) and sulfonation at different times (B) by using the current PAPS regeneration system and the AST IV-based co-factor recycling system.

References

1. Zhou Z, *et al.* A microbial-enzymatic strategy for producing chondroitin sulfate glycosaminoglycans. *Biotechnol Bioeng.* 2018.115:1561-1570.
2. Xi X, Hu L, Huang H, Wang Y, Xu R, Du G, Chen J, Kang Z. Improvement of the stability and catalytic efficiency of heparan sulfate *N*-sulfotransferase for preparing *N*-sulfated heparosan. *J Ind Microbiol Biotechnol.* 2023 17;50(1): kuad012.
3. Xu R, *et al.* Closed-loop system driven by ADP Phosphorylation from pyrophosphate affords equimolar transformation of ATP to 3'-phosphoadenosine-5'-phosphosulfate. *ACS Catal.* 2021. 11:10405-10415.
4. Cai T, *et al.* Cell-free chemoenzymatic starch synthesis from carbon dioxide. *Science.* 2021. 373:1523-1527.
5. Tavanti *et al.* ATP regeneration by a single polyphosphate kinase powers multigram-scale aldehyde synthesis *in vitro.* *Green Chem.* 2021. 23: 828-837.
6. Petchey *et al.* Biocatalytic synthesis of moclobemide using the amide bond synthetase McbA coupled with an ATP recycling system. *ACS Catal.* 2020. 10: 4659-4663.
7. Wang *et al.* Construction of a protein crystalline inclusion-based enzyme immobilization system for biosynthesis of PAPS from ATP and sulfate. *ACS Synth Biol.* 2023. 12:1487-1496.

“2: Several critical sulfotransferases involved in ULMW heparin synthesis rely on the presence of divalent cations such as Ca²⁺, Mn²⁺, and Mg²⁺ at concentrations ranging from 2 to 10 mM. It is crucial to exclude any phosphate from the reaction mixture when using divalent cations, especially Ca²⁺ and Mn²⁺. Including 2 mM polyP₆ in the reaction mixture results in a comparable amount of phosphate groups from about 10 mM phosphate solution, which is high enough to reduce the concentration of free divalent cations. Consequently, the PAPS regeneration system that incorporates polyP₆ cannot be effectively combined with these sulfation reactions, significantly limiting its practicality. Unfortunately, the authors did not provide a satisfactory explanation or

solution to address this limitation.”

Author’s response:

We agree that divalent cations such as Ca^{2+} or Mg^{2+} are crucial to maintain high activities of sulfotransferases. However, we coupled PAPS regeneration and sulfonation in our systems. Thus, with the consumption of polyP_6 , PO_4^{3-} was produced while PAPS was simultaneously regenerated for sulfonation. Once polyP_6 is consumed up, the sulfonation reaction tends to be finished. We didn’t find any obvious inhibitory effect that caused by the produced PO_4^{3-} . Moreover, the concentration of free divalent cations can be increased by external supplementation to thoroughly eliminate the possible inhibition from PO_4^{3-} . In fact, phosphate buffers have been widely used in enzyme-catalyzed buffer systems with addition of these divalent cations (*ACS Catal.* 2022. 12: 8372–8379, 50 mM PBS, 10mM MgSO_4), (*Nat Catal.* 2021. 4: 105–115. 200 mM MOPS/KOH (pH 7.5), 10 mM MgCl_2 , 40 mM polyphosphate, 100 mM NaHCO_3). Taken together, we think that the problem of phosphate incompatibility with metal ions in the buffer is not a severe problem and can be solved.

References

1. Wei Q, et al. Multienzyme assembly on caveolar membranes *in cellulo*. *ACS Catal.* 2022. 12: 8372-8379.
2. Scheffen M, et al. A new-to-nature carboxylation module to improve natural and synthetic CO_2 fixation. *Nat. Catal.* 2021. 4:105-115.

3: The authors employed five enzymes to perform the catalytic tasks a single enzyme could achieve. However, they did not provide a compelling side-by-side comparison demonstrating that when combined as a catalytic unit, the five enzymes offer superior accessibility and higher catalytic efficiency than the single enzyme (AST IV). For instance, they did not compare the yield and cost of sulfation by substituting AST IV in the No. 1 reaction with the five enzymes utilized in the No. 3 reaction. This lack of comparison raises questions about the advantages and effectiveness of employing multiple enzymes in this context.

In sulfation reactions utilizing AST IV for PAPS recycling, the addition of a catalytic amount of PAPS is adequate instead of using expensive PAP (refer to Table S3, reaction No. 1). Once the sulfo group of PAPS is transferred to the substrate, PAP is naturally

generated. Moreover, recent studies have addressed the cost of PAPS. Therefore, the cost calculation related to PAP may not be valid (refer to Table S3).

Author's response:

As the reviewer suggested, we have added the results in supplementary material. Also, to avoid reusing data, we modified Table S3 and only retained data from references. We agree that even PAPS regeneration system produced higher sulfonation value comparing to the AST IV-dependent regeneration system, four enzymes including one fused enzyme were recruited as the reviewer mentioned. Thus, a novel APS regeneration system by coupling specific sulfotransferase (three enzymes in total) was further proposed and constructed after discovering APS is also a sulfonate group donor. According to the suggestions from other three reviewers, we designed and constructed a rapid APS (adenosine 5'-phosphosulfate)-mediated sulfotransferase activity indicating system for screening sulfotransferase variants with high utilization efficiency towards APS. Eventually, we engineered an efficient APS regeneration system with short route, which achieved similar sulfonation efficiency comparing to the PAPS regeneration system. In our opinion this was the most important result in this manuscript, which has also been recognized by other three reviewers. Honestly, the developed APS regeneration system for sulfonation should be more applicable for practical applications.

Fig. 2. The comparison of three sulfonate donor regeneration systems: AST IV (two enzymes), PAPS (four enzymes) and APS (three enzymes).

In addition, we agree reviewer that in the AST IV-dependent system, PNPS and PAPS (but not PAP) were used as substrates while in the PAPS and APS regeneration systems, SO_4^{2-} , ATP and polyP₆ were used as substrates. In consideration of the difficulty for comparing the expression of different enzymes and the price for corresponding purification process, it was more reasonable to compare the price substrates, the sulfonation efficiency and the applicability for applications. Thus, we have modified some descriptions in the revised manuscript.

“4: The high density of negative charges in PolyP₆ presents a considerable obstacle not only for the purification of PAPS but also for the purification of any sulfated products obtained through the current PAPS regeneration system. PolyP₆ is expected to cause significant interference during the purification process, particularly when utilizing anion exchange resins, which are commonly employed in sulfo modification.”

Author’s response: We agree that polyP₆ possess high density of negative charges. However, in the PAPS regeneration system, with the proceeding of PAPS regeneration and sulfonation, polyP₆ was consumed up and transformed to PO_4^{3-} . Thus, the presence of high concentration of polyP₆ in the reaction system was invalid. In our previous study for PAPS biosynthesis, polyP₆ has been applied while no interference was observed (Xu *et al. ACS Catal.* 2021. 11:10405–415). More recently, we also achieved the purification of chondroitin sulfate in a phosphate- and sulfate-containing system (please see Fig. 3 below) (Zhang W *et al. Front Bioeng Biotechnol.* 2022. 10:951740). To further address the reviewer's concern about purification process, we purified trehalose-2-sulfate with the current PAPS regeneration system (please see Fig 4 below), and added this result in Supplementary Fig. 6. All these results suggest that the presence of phosphate has no significant effects on purification of the sulfated compounds.

In fact, polyP₆ or polyP_n (even longer than six) has been applied for *in vitro* enzymatic biosynthesis of hyaluronan (Gottschalk *et al. ChemCatChem* 2021. 13:1–11; Gottschalk *et al. ChemSusChem* 2022. 15, e202101071) and other products (aldehyde, moclobemide and starch). After finishing reaction, the polysaccharides such as hyaluronan, chondroitin sulfate or heparin were generally purified with alcohol precipitation. Also, small molecules such as PO_4^{3-} could also be easily removed by ultrafiltration. As a consequence, the application of polyP₆ for driving ATP regeneration for multiple enzyme reactions is not an issue. We hope the above results and answers can address reviewer's concerns.

Fig. 3. The CHS2 was purified with anion exchange chromatography (HiTrap 16/10 Q FF column). The depolymerized product was eluted on ÄKTA pure chromatography system with buffer A (20 mM phosphatebuffered saline, pH 8), and buffer B (20 mM phosphatebuffered saline containing 1000 mM NaCl, pH 8) at a flow rate of 3 mL/min. (A) Schematic diagram of sulfation-modification system for CSA biosynthesis. (B) Anion-exchange chromatograms of depolymerization products linearly eluted with 0–1000 mmol/L NaCl on a Q HP column at a flow rate of 3 mL/min. (C) MS spectra of CHS2 (Zhang W et al. *Front Bioeng Biotechnol.* 2022. 10:951740).

Fig. 4. The trehalose-2-sulfate was purified with anion exchange chromatography (HiTrap 16/10 Q FF column). The product was eluted on ÄKTA pure chromatography system with buffer A (20 mM Tris-HCl, pH 8), and buffer B (20 mM Tris-HCl containing 1000 mM NaCl) at a flow rate of 3 mL/min. (A) Anion-exchange chromatograms of depolymerization products linearly eluted with 0–1000 mmol/L NaCl on a Q HP column. (B) LC-MS spectra of trehalose-2-sulfate.

References

1. Xu R, et al. Closed-Loop System Driven by ADP Phosphorylation from pyrophosphate affords equimolar transformation of ATP to 3'-phosphoadenosine-5'-phosphosulfate. *ACS Catal.* 2021. 11: 10405-10415.
2. Zhang W, Xu R, Jin X, Wang Y, Hu L, Zhang T, Du G, Kang Z. Enzymatic production of chondroitin oligosaccharides and its sulfate derivatives. *Front Bioeng Biotechnol.* 2022. 10:951740.
3. Gottschalk et al. Integration of a nucleoside triphosphate regeneration system in the One-pot synthesis of UDP-sugars and hyaluronic acid. *ChemCatChem.* 2021. 13: 1-11.
4. Gottschalk et al. Repetitive synthesis of high-molecular-weight hyaluronic acid with immobilized enzyme cascades. *ChemSusChem* 2022. 15: e202101071.
5. Cai T, et al. Cell-free chemoenzymatic starch synthesis from carbon dioxide. *Science.* 2021. 373: 1523-1527.
6. Tavanti M, Hosford J, Lloyd RC, Brown MJB. ATP regeneration by a single polyphosphate kinase powers multigram-scale aldehyde synthesis *in vitro*. *Green Chem.* 2021. 23: 828-837.
7. Petchey MR, Rowlinson B, Lloyd RC, Fairlamb IJS, Grogan G. Biocatalytic synthesis of moclobemide using the amide bond synthetase McbA coupled with an ATP recycling system. *ACS Catal.* 2020. 10: 4659-4663.

Reviewers' Comments:

Reviewer #1:

Remarks to the Author:

The authors offered a detailed response to reviewers' comments and significantly improved the manuscript. The manuscript is worthy of being published in Nature Communications.

Reviewer #4:

Remarks to the Author:

1:

In this revised version, the authors have taken steps to address the concerns raised by the reviewers. While polyP_n has been widely utilized in polyphosphokinase-mediated ATP regeneration systems, it cannot be assumed that this ATP regeneration system can seamlessly integrate with the sulfated modification system. Hence, it is imperative to meticulously assess the coupling efficiency of these two systems.

To compare the sulfation efficiency between the current PAPS regeneration system and the AST IV-based co-factor recycling system, the authors employed HPLC/MS for quantifying trehalose-2-sulfate. However, there appears to be a notable difference between Supplementary Fig. 4B and Fig. 1A (in the response letter)/Supplementary Fig. 5. Please provide a detailed explanation for this discrepancy in the Methods section. Additionally, when sulfating trehalose using the AST IV system, the authors need to include an analysis diagram akin to Supplementary Fig. 4B for better clarity and comparative assessment.

"2:

While the authors have mentioned the coexistence of PBS and Mg²⁺ in the same reaction system, they have not presented a specific example demonstrating the applicability of the PAPS regeneration system containing polyP₆ to a reaction mixture that includes Ca²⁺ and Mn²⁺.

The authors should enhance the clarity of the article by either providing a detailed clarification or presenting experimental evidence to substantiate their assertion that the issue of phosphate incompatibility, specifically concerning polyP₆ and phosphate, with metal ions in the buffer, particularly Ca²⁺ and Mn²⁺, is not a significant concern and has been successfully addressed within their research. This step will contribute to a more robust scientific understanding of the viability and potential limitations of their proposed PAPS regeneration system in various reaction scenarios.

3:

Some enzymes may indeed have lower yields or activities. In the author's reaction system, the enzyme amount is substantial at 0.5 mg/mL. Therefore, in addition to comparing substrate costs and sulfonation efficiency, it would be valuable for the authors to provide information on enzyme yield, such as in mg/L or in terms of how many milligrams of total enzyme protein are required to synthesize 1 gram of the product. This data will assist readers in assessing the practical feasibility and efficiency of their enzymatic processes in terms of enzyme consumption and cost-effectiveness.

"4:

The author's response presents some confusion regarding the statement made in their initial reply,

where they indicated that polyP6 affects the purification of PAPS. In this subsequent response, they have used Supplementary Fig. 6 to demonstrate that trehalose-2-sulfate (which has weaker binding ability than PAPS) is not influenced by polyP6/phosphate in the reaction system. This contradiction requires clarification.

Additionally, Supplementary Fig. 6 displays two distinct UV absorption peaks for trehalose and trehalose-2-sulfate. Please confirm whether trehalose and trehalose-2-sulfate were quantified using an absorbance measurement at 218 nm.

Reviewer #1 (Remarks to the Author):

The authors offered a detailed response to reviewers' comments and significantly improved the manuscript. The manuscript is worthy of being published in Nature Communications.

Response: We gratefully thank reviewers for giving the good suggestions to enable us to improve the manuscript.

Reviewer #4 (Remarks to the Author):

“1: In this revised version, the authors have taken steps to address the concerns raised by the reviewers. While polyP_n has been widely utilized in polyphosphokinase-mediated ATP regeneration systems, it cannot be assumed that this ATP regeneration system can seamlessly integrate with the sulfated modification system. Hence, it is imperative to meticulously assess the coupling efficiency of these two systems.

Response: We appreciate the reviewers for these valuable comments. Accordingly, we have conducted more experiments and modified some descriptions in the revised manuscript. The new results have been added in the Supplementary Information.

In our previous studies, we have achieved the active expression of sulfotransferase C4ST and the biosynthesis of chondroitin sulfate A with the AST IV-based PAPS regeneration system. In view of the compatibility for future *in vivo* applications and to avoid the accumulation of toxic compound during reaction, especially after we constructed and optimized the PAPS synthesis module (II) in our previous study (Xu et al. *ACS Catal.*), a new PAPS regeneration system was constructed. To this end, two other modules: the module I for ATP regeneration and the module III for AMP regeneration were designed and the key enzymes for ATP regeneration (Fig. 1C, D) and the conversion of PAP to AMP (Fig. 1E, F) were individually screened and characterized. Then, we introduced the corresponding sulfotransferases and coupled

these modules step by step (please see Fig. 2, module I to II to III; Supplementary Fig. 1C, module I to II; Supplementary Fig. 2, module II to III; Supplementary Fig. 4D, module III to I). The results proved that three modules are able to work well together.

In our opinion, the AST IV dependent method is suitable for the rapid characterization of enzyme activities of sulfotransferases and the preparation of small samples for functional assessment. In comparison, as an alternative, the constructed new PAPS regeneration system is suitable for multienzyme catalysis of products *in vivo* or *in vitro* because of its compatibility and biosafety.

References

Xu R, Wang Y, Huang H, Jin X, Li J, Du G, and Kang Z. Closed-loop system driven by ADP phosphorylation from pyrophosphate affords equimolar transformation of ATP to 3'-phosphoadenosine-5'-phosphosulfate. *ACS Catal.* 2021. 16:10405–10415.

“To compare the sulfation efficiency between the current PAPS regeneration system and the AST IV-based co-factor recycling system, the authors employed HPLC/MS for quantifying trehalose-2-sulfate. However, there appears to be a notable difference between Supplementary Fig. 4B and Fig. 1A (in the response letter)/Supplementary Fig. 5. Please provide a detailed explanation for this discrepancy in the Methods section. Additionally, when sulfating trehalose using the AST IV system, the authors need to include an analysis diagram akin to Supplementary Fig. 4B for better clarity and comparative assessment.”

Response: We thank the reviewer for great attention to every detail. In this study, we employed HPLC/MS to quantify sulfated compounds. First, the **total ion chromatography** (parent ion fragments) was characterized and shown in supplementary Fig. 4B, which was used for evaluating the separation of the products and substrates and identifying whether Stf0 has enzyme activity.

To quantify the sulfated products, we isolated the mass spectrum corresponding to the m/z values of the designated products from the **total ion chromatography** (parent ion

fragments). Subsequently, the concentrations of sulfated products were calculated based on the area of the selected peak. (Fig. 2C, 2E, Fig. 4 and Fig. 1A in the previous response letter/Supplementary Fig. 5B). As the reviewer suggested, we have added detailed explanations in the Methods section. Also, the corresponding total ion chromatography analysis of the first mass spectrometry for trehalose-2-sulfate with the AST IV-mediated reaction system was added (Supplementary Fig. 5A). Please see below.

Separation and identification of trehalose and trehalose-2-sulfate using total ion chromatography (A) and quantification of trehalose-2-sulfate using secondary mass spectrometry (B)

“2: While the authors have mentioned the coexistence of PBS and Mg^{2+} in the same reaction system, they have not presented a specific example demonstrating the applicability of the PAPS regeneration system containing polyP₆ to a reaction mixture that includes Ca^{2+} and Mn^{2+} . The authors should enhance the clarity of the article by either providing a detailed clarification or presenting experimental evidence to substantiate their assertion that the issue of phosphate incompatibility, specifically concerning polyP₆ and phosphate, with metal ions in the buffer, particularly Ca^{2+} and Mn^{2+} , is not a significant concern and has been successfully addressed within their research. This step will contribute to a more robust scientific understanding of the viability and potential limitations of their proposed PAPS regeneration system in various reaction scenarios.”

Response: Thank the reviewer for this rigorous comment. In our previous response, we provided some references to prove that the coexistence of polyP_n and Mg²⁺ in the same reaction system can not affect the progress of catalytic reactions. As the reviewer commented, we have conducted the experiments and investigated the effects of supplementary of extra metal ions Mg²⁺, Ca²⁺ and Mn²⁺ on the PAPS regeneration system. The results were shown below. When Ca²⁺ was added, the production of sulfated trehalose slightly dropped while *N*-sulfated heparosan was improved. When Mn²⁺ was added, both sulfated trehalose and *N*-sulfated heparosan were improved in varying degrees. According to these results, it could be concluded that these commonly used metal ions Mg²⁺, Ca²⁺ and Mn²⁺ were not affected apparently by the polyP_n. These new results demonstrated that the constructed PAPS regeneration system should be suitable for the reaction scenarios with these metal ions.

Influence of Ca²⁺ and Mn²⁺ on PAPS regeneration system coupled with trehalose-2-sulfate production (A) or *N*-sulfated heparosan production (B)

“3: Some enzymes may indeed have lower yields or activities. In the author's reaction system, the enzyme amount is substantial at 0.5 mg/mL. Therefore, in addition to comparing substrate costs and sulfonation efficiency, it would be valuable for the authors to provide information on enzyme yield, such as in mg/L or in terms of how many milligrams of total enzyme protein are required to synthesize 1 gram of the product. This data will assist readers in assessing the practical feasibility and efficiency of their enzymatic processes in terms of enzyme consumption and cost-effectiveness.

Response: We appreciate the reviewer for this good suggestion. Accordingly, we have provided more information on yields of enzymes involved in PAPS regeneration system and APS regeneration system in the supplementary information. These data will facilitate readers to optimize the reaction system towards specific sulfated products. The strains were all cultivated in TB medium at designated temperature, purified via Ni-NTA columns and quantified via a modified Bradford protein assay kit (Sangon, China) using bovine serum albumin as the standard.

Enzymes	Cultivation conditions	OD ₆₀₀	Expression level (g/L)
ASAK	30°C, 15 h	13	0.38 ± 0.03
KpCysQ	30°C, 15 h	13	0.62 ± 0.04
PPK2 ^{s,c}	30°C, 15 h	14	0.51 ± 0.03
ATPS ^s	30°C, 15 h	13	0.32 ± 0.02
AST IV ¹	16°C, 24 h	8	0.26 ± 0.02

Reference

1. Zhou, Z, Li, Q, Huang H, Wang H, Wang Y, Du G, Chen J, Kang Z. A microbial-enzymatic strategy for producing chondroitin sulfate glycosaminoglycans. *Biotechnol Bioeng*. 2018. 115:1561-1570.

“4: The author's response presents some confusion regarding the statement made in their initial reply, where they indicated that polyP₆ affects the purification of PAPS. In this subsequent response, they have used Supplementary Fig. 6 to demonstrate that trehalose-2-sulfate (which has weaker binding ability than PAPS) is not influenced by polyP₆/phosphate in the reaction system. This contradiction requires clarification. Additionally, Supplementary Fig. 6 displays two distinct UV absorption peaks for trehalose and trehalose-2-sulfate. Please confirm whether trehalose and trehalose-2-sulfate were quantified using an absorbance measurement at 218 nm.

Response: We are sorry for the ambiguous description in our last responses. We agree that the presence of polyP₆ should have influence on PAPS purification because of the high density of negative charges. In our previous study, we successfully purified PAPS in the reaction system containing pyrophosphate (PPi). Thus, we speculated that PPi with short chain length should not generate significant effect on PAPS purification. In contrast, herein, we coupled PAPS regeneration and **sulfonation** in a common system. Although polyP₆ was added initially, polyP₆ was continuously consumed with the sulfonation reaction progressed. Once the sulfonation reaction was finished, polyP₆ would be consumed up. Therefore, the downstream purification of the sulfated compounds would not be affected by polyP₆.

For the second comment, the ultraviolet detector was used in both separation and preparation of the sulfated compounds. To separate trehalose and trehalose-2-sulfate, we verified the strong absorbance at the ultraviolet region of 218 nm (A, below) according to wavelength scanning and a good linear relationship between concentration of trehalose and absorbance value (B, below). Thus, the UV detector at 218 nm was applied for separation and preparation of trehalose and trehalose-2-sulfate. After purification, the HPLC/MS was employed for quantifying purified trehalose-2-sulfate.

Wavelength scanning of trehalose (A); Standard curve line of concentration of trehalose and absorbance value at 218 nm (B).

Reviewers' Comments:

Reviewer #4:

Remarks to the Author:

I have no further queries for the authors.

In order to enhance the comprehensiveness and gravity of the article, the author should include all tables and diagrams from "Response to Referees Letter" in supplementary materials.

Reviewer #4 (Remarks to the Author):

I have no further queries for the authors.

In order to enhance the comprehensiveness and gravity of the article, the author should include all tables and diagrams from "Response to Referees Letter" in supplementary materials.

Response: We gratefully thank reviewers for giving the good suggestions to enable us to improve the manuscript.

As suggested, we have included the required documents in the Supplementary Information and compiled the previous responses to reviewers to bolster the publication of the manuscript. We would like to extend our sincere gratitude for the diligent efforts of the reviewers.